



# A nonhydrostatic oceanic regional model ORCTM v1 for
# internal solitary wave simulation
Hao Huang[1], Pengyang Song[1,2], Shi Qiu[1], Jiaqi Guo[1], Xueen Chen[1]
[1]College of Oceanic and Atmospheric Sciences, Ocean University of China, Qingdao, 266100, China
[2]Alfred Wegener Institute for Polar and Marine Research, Bremerhaven, Germany
*Correspondence to*: Xueen Chen (xchen@ouc.edu.cn)
**Abstract.**
An Oceanic Regional Circulation and Tide Model (ORCTM version 1) including the nonhydrostatic
dynamics module which can numerically reproduce the Internal Solitary Waves (ISWs) dynamics, is
presented in this paper. The model open boundary conditions are also supportive of regional baroclinic
tidal wave simulations.
The incompressible Boussinesq equations in z-coordinates consider the three-dimensional and fully
nonlinear forms. The pressure field is also decomposed into the surface, hydrostatic and nonhydrostatic
components on the orthogonal curvilinear Arakawa-C grid. The nonhydrostatic pressure determined by
the intermediate velocity divergence field is obtained via solving a three-dimensional Poisson equation
based on a pressure correction method. Model validation experiments for ISWs simulations with the
topographic change in the two-layer and continuously stratified ocean demonstrate that the ORCTM has
a considerable capacity for reproducing the life cycle of Nonlinear Internal Waves evolution and tide-
topography interactions.
**1. Introduction**
Internal Waves (also called Internal Gravity Waves) activities have been observed frequently across
the stratified ocean and play a significant role in the multiscale energy cascade (Mtfller, 1976).
Observations reveal that the Internal Waves, especially the high-frequency Internal Solitary Waves, could
contain significant potential energy with strong vertical shear, mixing, and wave breaking, leading to a
dramatic change of the currents and density structures (Ramp et al., 2004; Vlasenko et al., 2010; Huang
et al., 2016), violent overturning bringing sediment and nutrient from the seafloor to the surface (Wang
et al., 2007), even irretrievable damages to some underwater vehicles (Duda et al., 2006) and deep-water



drilling (Osborne et al., 1978). Basically, astronomical tides passing the abrupt topography can cause the
generation of the baroclinic tides (also called internal tides) with multi-modal structures then capable of
propagation, disintegration, and dissipation in the ocean (Vlasenko et al., 2005; 2010). The low-mode of
baroclinic tides can travel thousands of kilometers with the long horizontal wavelengths about ten of
kilometers (Baines, 1982). Furthermore, the inclusion of nonlinear and nonhydrostatic effects permits
the evolution of the Nonlinear Internal Waves (hereafter NIWs) even the Internal Solitary Waves
(hereafter ISWs) derived from the steepening of low-mode internal tides as the consequence of the ever-
changing terrain and background stratification (Gerkema and Zimmerman, 1995; legg and Adroft, 2003).

Numerical Ocean models are one of the most effective tools to study Internal Waves compared to

theoretical methods, in-situ observations, and laboratory investigations. Specifically, the ocean models
with the hydrostatic balance approximation have been used to explore the circulation and tide processes
across the stratified ocean thanks to their excellent performance with fairly high accuracy (Marshall et
al., 1997b; Chen et al., 2003; Shchepetkin and McWilliams, 2005; Ko et al., 2008). However, the high-
frequency nonlinear ISWs and the steepening of the internal tides cannot be depicted by the hydrostatic
models, because the strong vertical current with its order of magnitude equals the horizontal one via the
scale analysis method (Marshall et al., 1997a), when the three-dimensional Navier-Stokers equations
should be considered thoroughly. In other words, hydrostatic approximation due to omitting other terms
in the vertical momentum equation results in the inapplicability of the nonhydrostatic dynamics (Lai et
al., 2010). Thus, it is indispensable for simulating the nonlinear and large amplitude ISWs to develop a
nonhydrostatic ocean model and consider nonhydrostatic dynamics.

A robust ocean model considering the nonhydrostatic dynamics should satisfy two requirements

synchronously at least: 1) The high enough accuracy of meso-to-big scales simulation must be under
guarantee, such as the large-scale wind-induced circulation and mesoscale eddies reconstructed and
mainly influenced by the hydrostatic balance; 2) Meanwhile, it is the concerned small-meso scales with
the higher spatial and temporal resolution that are resolved finely under the nonhydrostatic balance, for
example, there is the simulation being able to describe the cradle-to-grave process for the tide-topography
interactions, the dispersive effects and nonlinear steepening of the baroclinic tides, and the breaking and
dissipation of strong nonlinear ISWs. The nonhydrostatic simulation can apply to the small-to-big scales
across the stratified ocean simultaneously, which can be assumed as one of the directions for research
and development of the nonhydrostatic ocean model. In reality, there have been some nonhydrostatic

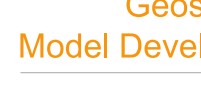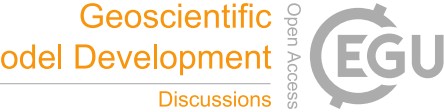

ocean models or ones considering nonhydrostatic dynamics coming out in the past years, such as
MITgcm (Marshall et al., 1997a;1997b;1998), SUNTANS (Fringer et al., 2006), and ROMS (Kanarska
et al., 2007). All above have been used to realize a series of two or three-dimensional nonhydrostatic
numerical studies, including the instability of small-scale flows in the laboratory experiment (Lai et al.,
2010; Li et al., 2022), Internal Solitary Waves in the continental shelves (Vlasenko et al., 2010, Zeng et
al., 2019) and the hydraulic Lee wave around the seamount (Kanarska et al., 2007; Liu et al., 2016), etc.
Nevertheless, there is still no widespread use for the nonhydrostatic ocean model due to costly
computational resources. The corresponding nonhydrostatic solution to an extensive sparse linear
equation system is too difficult to solve directly for the 3-D oceanic environment, which usually demands
of large amounts of iteration times, fast convergent speed, and PC storage occupation. In addition, the
different kinds of sub-grid parameterization schemes have a profound impact on the model results with
a necessity for appropriate one to be assessed, and most of these model codes are seldom shared or of
open source. Suppose we develop a nonhydrostatic ocean model based on an original hydrostatic
framework model. In that case, the nonhydrostatic module will include the vertical momentum equation
with some terms related to the vertical velocity simultaneously complemented in the horizontal
momentum equation. Besides, based on the idea of the fractional step method (Press et al., 1988;
Armfield and Street, 2002), the total pressure is to be decomposed into hydrostatics and nonhydrostatic
components (Marshall et al., 1997a; Lai et al., 2010). The former corresponds to the result of hydrostatic
balance, and the divergence for intermediate velocity limits the latter to correct the local velocity fields
called the "pressure correction" method (Stansby and Zhou, 1998; Fringer et al., 2006; Kanarska et al.,
2007; Lai et al., 2010). With these methods, the nonhydrostatic dynamics simulation can be fulfilled
economically comparatively in harmony with the original physical framework as an extension of the
hydrostatic ocean model.

In this context, we have implemented the nonhydrostatic dynamic algorithm into an Oceanic

Regional Circulation and Tide Model (hereafter ORCTM) and demonstrated its ability and performance
of reproducing the life cycle of ISWs and NIWs in different hydrodynamic environments. The rest of the
paper is organized as follows. In Section 2, the basic framework of the ORCTM including control
equations, open boundary conditions, and nonhydrostatic algorithms is described. In Section 3, a series
of numerical validation experiments results are presented, aiming at the simulation of the overall
processes of the ISWs. In the last section, we come to conclusions and prospect the directions.





**2. Model development**

Derived from the basic framework of the Max-Planck-Institute ocean model featuring the global

hydrostatic modelling with the staggered Arakawa-C grid and the cartesian coordinates system

supporting the orthogonal curvilinear form (Marsland et al., 2003; Chen et al., 2005), in this paper, the

latest Max-Planck-Institute ocean model has been upgraded to realize the circulation and tide simulation

with the open boundary conditions for regional ocean modelling, and the z-level grid applied has the

partial filled cell capability to adjust the distance of the vertical grid on seabed for fitting into the realistic

terrain, which will be referred to hereafter as ORCTM. It is under the laws of the Boussinesq, rotating

and fully nonlinear Navier–Stokes fluid that the ORCTM can be used to study both the hydrostatic and

nonhydrostatic ocean dynamics, which namely can reproduce the nonhydrostatic dynamic processes such

as upwelling and downwelling, NIWs, and ISWs, etc.

**2.1. Control Equations**

The primitive three-dimensional control equations include the momentum, continuity, free surface,

potential temperature, salinity, and density equations given as follows.

$$\frac{\partial u}{\partial t} + u\frac{\partial u}{\partial x} + v\frac{\partial u}{\partial y} + w\frac{\partial u}{\partial z} - fv + \tilde{f}w = -\frac{1}{\rho_c}\frac{\partial P}{\partial x} - g\frac{\partial \varsigma}{\partial x} + F_{Vx} + F_{Hx} + \mathcal{F}_x \tag{1}$$

$$\frac{\partial v}{\partial t} + u\frac{\partial v}{\partial x} + v\frac{\partial v}{\partial y} + w\frac{\partial v}{\partial z} + fu = -\frac{1}{\rho_c}\frac{\partial P}{\partial y} - g\frac{\partial \varsigma}{\partial y} + F_{Vy} + F_{Hy} + \mathcal{F}_y \tag{2}$$

$$\frac{\partial w}{\partial t} + u\frac{\partial w}{\partial x} + v\frac{\partial w}{\partial y} + w\frac{\partial w}{\partial z} - \tilde{f}u = -\frac{1}{\rho_c}\frac{\partial P}{\partial z} - g + F_{Vz} + F_{Hz} + \mathcal{F}_z \tag{3}$$

$$\frac{\partial u}{\partial x} + \frac{\partial v}{\partial y} + \frac{\partial w}{\partial z} = 0 \tag{4}$$

$$\frac{\partial \varsigma}{\partial t} = -\nabla_h \cdot \int_{-H}^{\varsigma} \boldsymbol{u}_H dz \tag{5}$$

$$\frac{\partial \theta}{\partial t} + u\frac{\partial \theta}{\partial x} + v\frac{\partial \theta}{\partial y} + w\frac{\partial \theta}{\partial z} = F_{V\theta} + F_{H\theta} + Q_\theta \tag{6}$$

$$\frac{\partial S}{\partial t} + u\frac{\partial S}{\partial x} + v\frac{\partial S}{\partial y} + w\frac{\partial S}{\partial z} = F_{VS} + F_{HS} + Q_s \tag{7}$$

$$\rho = \rho(\theta, S, P) \tag{8}$$

In the cartesian coordinates system, $t$ is the time; $\partial/\partial t$ is the time partial derivative; $x, y$ and $z$

axes direct eastward, northward, and upward respectively; The horizontal velocity vector is $\boldsymbol{u_h} = (u, v)$;

$w$ is the vertical velocity; $\varsigma$ is the change of the free surface elevation; $P, \theta$ and $S$ are pressure,

potential temperature and salinity; $\rho_c$ is the reference density of sea water under the Boussinesq





approximation; The first and second Coriolis parameters are $f = 2\Omega \sin \varphi$ and $\tilde{f} = 2\Omega \cos \varphi$, where $\Omega$
is the rotational angular speed and $\varphi$ is the geographic latitude. $\nabla_H$ is the horizontal divergence
operator; $Q_\theta$ and $Q_s$ are source or sink terms about potential temperature and salinity. The equation of
seawater state is the polynomial form for the density $\rho$ advocated by the Joint Panel on Oceanographic
Tables and Standards (Fofanoff and Millard, 1983).

The additional forcing term vector $\boldsymbol{\mathcal{F}} = (\mathcal{F}_x, \mathcal{F}_y, \mathcal{F}_y)$ can consider tidal potential forcing, river

runoff, and open boundary outflow and inflow. The horizontal eddy viscosity vector is $\boldsymbol{F}_H =$
$(F_{Hx}, F_{Hy}, F_{Hy})$ described with the scale-dependent biharmonic formulation (Wolff et al., 1997;
Marsland et al., 2003), and the horizontal diffusivity terms of temperature and salinity are $F_{H\theta}$ and
$F_{HS}$ supporting the harmonic forms. Besides, the vertical eddy viscosity vector is $\boldsymbol{F}_V = (F_{Vx}, F_{Vy}, F_{Vy})$
and eddy diffusivity terms are $F_{V\theta}$ and $F_{VS}$. Here, the vertical eddy turbulent frictions which both are
specified to depend on the Richardson number $Ri$ via the modified PP81 parameterization scheme
(Pacanowski and Philander, 1981). The viscous terms all above are expressed as

$$\boldsymbol{F}_H = -\nabla_h \cdot (B_H \nabla_h \Delta \boldsymbol{u}), \quad \boldsymbol{F}_V = \frac{\partial}{\partial z}\left(A_V \frac{\partial \boldsymbol{u}}{\partial z}\right) \tag{9}$$

$$F_{\gamma H} = D_H \Delta \gamma, \quad F_{\gamma V} = \frac{\partial}{\partial z}\left(D_V \frac{\partial \gamma}{\partial z}\right), \qquad \gamma = \theta, S \tag{10}$$

$$A_V^{n+1} = (1 - \lambda)A_V^n + \lambda(A_{V0}(1 + \alpha \cdot Ri)^{-2} + A_w + A_b) \tag{11}$$

$$D_V^{n+1} = (1 - \lambda)D_V^n + \lambda(D_{V0}(1 + \alpha \cdot Ri)^{-3} + D_w + D_b) \tag{12}$$

$$Ri = \frac{N(z)^2}{(\partial u/\partial z)^2 + (\partial v/\partial z)^2} \tag{13}$$

where $\Delta = \nabla_h \cdot \nabla_h$ is the horizontal Laplace operator; $B_H$ and $D_H$ are parameterized with the horizontal
grid resolution; $N(z)$ is the buoyancy frequency. $A_V^{n+1}$ and $D_V^{n+1}$ are updated on formulas (11) and (12)
with the time relaxation coefficient $\lambda$ at every timestep. Apart from the background viscous coefficients
$A_b$ and $D_b$ due to internal waves breaking, the modified PP81 scheme also considers the wind-induced
turbulent coefficients $A_w$ and $D_w$ associated with the local mixed layer depth and 10m wind speed
(Marsland et al., 2003). Here, the constant number $\alpha$ is set to be 5. And the adjustable parameters $A_{V0}$
and $D_{V0}$ can be determined by estimating energy flux at every grid point. As for the boundary condition,
the slip conditions are specified at surface and bottom boundaries where the wind stress $\boldsymbol{\tau}_w$ is based on
the input, and the bottom drags $\boldsymbol{\tau}_b$ are described by linear and quadratic functions (Arbic and Scott, 2008).





### 2.2. Settings of Open Boundary Condition


It is fundamental for the regional model to be configured by an open boundary condition that avoids
reflection waves effectively so that the outward waves can freely flow through the boundaries.
Meanwhile, the external inputs such as the tidal waves can stably force the model domain through the
boundaries, satisfying the needs for consistency in hydrodynamics and computational mathematics. Here,
we use the relaxation boundary conditions with sponge layers consulting Zhang et al. (2011) that enable
to dampen the reflection waves in the model domain and refrain from the sharp gradients of water
properties caused by the prescribed boundary values on the boundaries. Specifically, we add a relaxation
term $M$ formularized with the exponential function to the right-hand side of momentum equations (1)
to (3), temperature equation (6), and salinity equation (7) expressed as

$$M(x,y,z,t) = -\left(\frac{m(x,y,z,t) - m_b(x,y,z,t)}{\tau}\right) \cdot e^{-\frac{4 \cdot r(x,y)}{L_{sp}}} \tag{14}$$

In this formula (14), $m_b$ is the boundary values of requisite model variables through the boundaries
including velocity, potential temperature, and salinity; $m$ is the corresponding relaxation result in the
interiors; $r$ is the distance from the boundary. Moreover, $\tau$ and $L_{sp}$ are artificially prescribed adjustment
parameters referring to the time-scale coefficient and the thickness of the sponge layers. The model target
variables over the sponge layer will relax exponentially to the boundary values through the relaxation
term, where relaxation is modulated by $\tau$ and $L_{sp}$ in the exponential shape. To restrain the reflection of
outflow current, $\tau$ and $L_{sp}$ need to be determined in advance via estimating the energy flux of internal
signals through the boundaries. This open boundary relaxation condition is suitable for the numerical
study of the large ISWs so that the strong, nonlinear, and nonhydrostatic wave and current signals will
dampen gradually.

### 2.3. Implement of Nonhydrostatic Algorithms


According to the momentum equations (1) to (3), the total pressure $P$ consists of sea surface
pressure $p_s$, hydrostatic pressure $p_h$, and nonhydrostatic pressure $p_{nh}$ given as follows.

$$P = p_s(x,y) + p_h(x,y,z) + p_{nh}(x,y,z) \tag{15}$$

$$-\frac{1}{\rho_c}\frac{\partial p_h}{\partial z} - g = 0 \tag{16}$$

Hydrostatic pressure $p_h$ can be calculated from the hydrostatic balance Eq. (16). It is negligible
for sea surface pressure term $p_s$ to impact on the water column if the external atmospheric forcing is





excluded. Furthermore, based on the idea of the fractional step method (Press et al., 1988; Kanarska et
al., 2007), the intermediate velocity field $\tilde{\boldsymbol{u}}$ will be updated via the nonhydrostatic pressure $p_{nh}^n$
gradients, which is defined and can be obtained via the Eqs. (17) to (19) discretized as follows.

$$\frac{\tilde{u} - u^n}{\Delta t} = -G_x - \frac{1}{\rho_c}\frac{\partial p_{nh}^n}{\partial x} \tag{17}$$

$$\frac{\tilde{v} - v^n}{\Delta t} = -G_y - \frac{1}{\rho_c}\frac{\partial p_{nh}^n}{\partial y} \tag{18}$$

$$\frac{\tilde{w} - w^n}{\Delta t} = -G_z - \frac{1}{\rho_c}\frac{\partial p_{nh}^n}{\partial z} \tag{19}$$

Where the superscript $n$ means the current timestep and the vector $\boldsymbol{G} = \left(G_x, G_y, G_y\right)$ represents the
advection terms, Coriolis terms, eddy viscosity terms, and hydrostatic pressure gradients terms.
Subsequently, the discretized partial equations (20) to (22) are established with the relationship between
the nonhydrostatic pressure perturbation $p_{nh}'$ gradients and the next timestep $n+1$ velocity field. Then
nonhydrostatic pressure at the next timestep $n+1$ is defined as (23) in the light of the pressure correction
method. To obtain nonhydrostatic pressure perturbation the continuity equation (4) needs to be
substituted into Eqs. (20) to (22) to eliminate the following timestep $n+1$ velocity field with the three-
dimensional Poisson equation (24), which demonstrates that the nonhydrostatic pressure depends on the
vanishes of the divergence-free velocity fields.

$$\frac{u^{n+1} - \tilde{u}}{\Delta t} = -\frac{1}{\rho_c}\frac{\partial p_{nh}'}{\partial x} \tag{20}$$

$$\frac{v^{n+1} - \tilde{v}}{\Delta t} = -\frac{1}{\rho_c}\frac{\partial p_{nh}'}{\partial y} \tag{21}$$

$$\frac{w^{n+1} - \tilde{w}}{\Delta t} = -\frac{1}{\rho_c}\frac{\partial p_{nh}'}{\partial z} \tag{22}$$

$$p_{nh}^{n+1} = p_{nh}^n + p_{nh}' \tag{23}$$

The Poisson equation (24) can be discretized into a linear matrix Eq. (25) where the right-hand side
$\boldsymbol{B}$ is determined by the divergence of the intermediate velocity field. The adjoint matrix $\boldsymbol{A}$ represents the
discrete three-dimensional Laplacian operator with a size of the number of model cells. The specific
discrete process for Eq. (24) is introduced in Appendix A.

$$\frac{\partial^2 p_{nh}'}{\partial x^2} + \frac{\partial^2 p_{nh}'}{\partial y^2} + \frac{\partial^2 p_{nh}'}{\partial z^2} = \frac{\rho_c}{\Delta t}\left(\frac{\partial \tilde{u}}{\partial x} + \frac{\partial \tilde{v}}{\partial y} + \frac{\partial \tilde{w}}{\partial z}\right) \tag{24}$$

$$\boldsymbol{A}p_{nh}' = \boldsymbol{B} \tag{25}$$

$$\nabla p_{nh}' \cdot \boldsymbol{n} = 0 \tag{26}$$





The proper boundary conditions need to be considered for solving this Poisson equation (24). Here,
at the solid boundaries the homogeneous Neumann boundary condition, also called the Zero-gradient
condition (26), is used with a good compatibility with the no flux normal to slope, where $\boldsymbol{n}$ is the normal
unit vector (Marshall et al., 1997a). Moreover, we assume that nonhydrostatic dynamic processes are
weak enough at the sea surface and open boundaries. In other words, the input signals through the
boundaries are dominantly hydrostatic with nonhydrostatic pressure perturbation close to zero. More
specifically, the nonhydrostatic dynamic framework is restricted in the interior. Hence the Zero-gradient
condition is also used at the open boundaries in case of sharp nonhydrostatic pressure gradients. With the
above boundary conditions this linear system (25) can be solved via the Krylov subspace method with
the PETSc's assistance on parallel computers under the standard MPI-based framework (Balay et al.,
2020). It is a highly efficient method devised to precondition the huge and sparse matrix $\boldsymbol{A}$. Here, the
multigrid preconditioner (Smith et al., 1996) and flexible generalized minimal residual algorithm (Saad,
1993) are employed in numerical validation experiments in this paper to reduce the computational costs.
**3. Model applications and assessments**
Here we present a series of ideal numerical validation experiments to explore the correctness and
compatibility of nonhydrostatic algorithms together with the ORCTM. These test cases range from the
laboratory-scale cases in a tank to field-scale cases like the northern South China Sea with open
boundaries in allusion to the dynamics of ISWs. The first case is the lock-exchange problem as the
preliminary validation. The second to fourth cases are designed to explore the nonlinear evolution of
ISW caused by its interactions with the changing terrain. The last one is the generated nonlinear internal
waves case in a double-ridge environment analogous to the Luzon Strait via boundary tidal forcing,
which aims at the generation and disintegration of NIWs to examine the effectivity about open boundary
module under the nonhydrostatic algorithms. All test experiments above can demonstrate the capabilities
of simulating the evolution of NIWs and ISWs with good compatibility with the nonhydrostatic
algorithms.
**3.1. The lock-exchange problem**
When the shear currents flow between the two different density fluids, the Kelvin-Helmholtz



instability (hereafter K-H instability) will appear to cause the turbulent diapycnal mixing (Lawrence et
al., 1991; Cushman-Roisin, 2005). The perturbation on the interface begins to develop and stimulate
gradually numerous eddies with the high wavenumber due to the dispersion. The order of vertical flow
magnitude is comparable to the horizontal one so that the nonhydrostatic effect matters from the
beginning to the end.

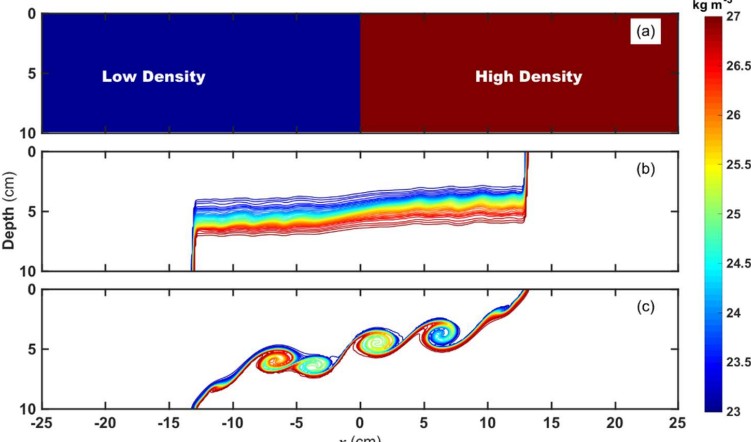

Figure 1. (a) The initial density $\rho'$ (hereafter the same expression) field of the lock-exchange case, and
their contour plots of density at $t = 4.5$ s where the contour interval is 0.1 kg m$^{-3}$ under the hydrostatic
(a) and nonhydrostatic (b) framework.
We set a rectangle tank separated by a vertical board in the middle at the $x$-axis origin. Both sides
of the tank are separately filled with two different density fluids in Fig. 1a. The gravitational adjustment
will proceed when the central board is disengaged just like a lock gate. Here, we refer to the previous
configuration as a 2-D problem (Härtel et al., 2000; Fringer et al., 2006; Lai et al., 2010). The horizontal
length $L$ is set to 50 cm and the static water height is 10 cm without the topographic change in the tank.
And the horizontal and vertical resolution are both 0.001 m. In order to reduce the dissipations out of
friction and close to the ideal status, several sensitivity experiments were explored so that the bottom
friction coefficient is set to zero and $A_{V0}$ and $D_{V0}$ in formulas (11) and (12) are both set to 2×10$^{-6}$ m$^2$
s$^{-1}$. The K-H instability process develops rapidly in this configuration with good eddies reconstruction
and outstanding waves breaking. Water density averages are calculated based on the prescribed salinity
difference on the left and right sides of the tank $\rho_l = 1023.05$ kg m$^{-3}$ and $\rho_r = 1026.95$ kg m$^{-3}$. The
same configuration experiment above but under the hydrostatic balance scheme is also run for



comparison. Figures 1b and 1c show the comparison results of density (define $\rho' = \rho - 1000$ kg m$^{-3}$) at
the same time under the hydrostatics and nonhydrostatic balance assumption, which proves that the K-
H instability cannot appear resulting from the inapplicability of the hydrostatics balance. The
perturbation on the density interface is so tiny that the density fronts cannot develop in the upper and
lower layer. To sum up, the mixing caused by the overturning and shear is too weak to be seen. On the
contrary, via the nonhydrostatic scheme, the numerous eddies can appear and dissipate energy from the
perturbation, mixing the high and low-density water on the interface vigorously. More specifically, the
energy is transmitted to the small-scale eddies across the density fronts due to dispersion and nonlinearity.

The evolution process of K-H instability is shown in Fig. 2. It is out of gravitational adjustment that

the density fronts with heavy water in the bottom and light one in the upper move to the left and right,
respectively, causing a velocity shear field and clockwise rotating interface in Fig. 2a. The shear strength
gradually increases until breaking the critical point of restoring force that depends on the density gradient.
A series of eddies then develop from the middle to both sides of the tank with the turbulent rolling and
overturning. They mix the water body with high density at the bottom and upper one with the low density,
forming numerous mixing areas in Figs. 2b and 2c. When the bottom density flow is reflected on the wall
on the left side, the whole adjustment process begins to develop in reverse of Figs. 2d and 2e, but the
strength of subsequent eddies is significantly weakened due to the energy dissipation.

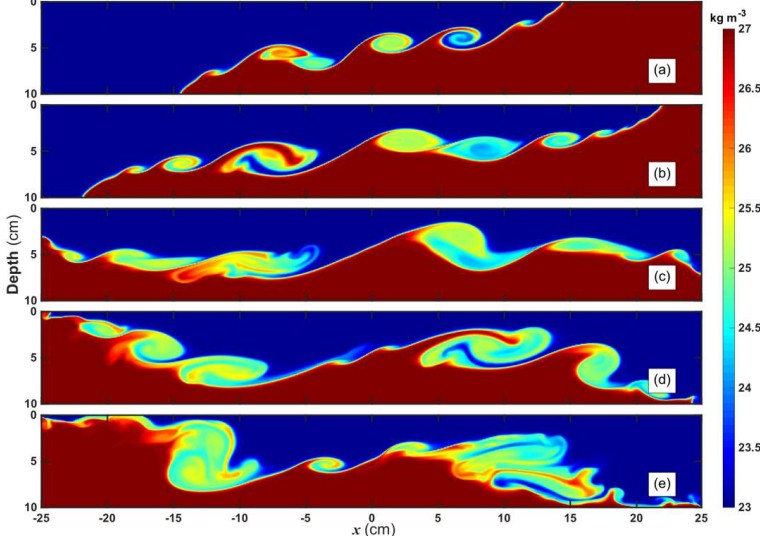

Figure 2. Density field evolution at $t$ = (a) 5.0, (b) 7.5, (c) 10.0, (d) 12.5, and (e) 15.0 s




From the density distribution, we can find that the density fronts and eddies processes accompany
the gravitational adjustment. In reality, the gravitational potential and kinematical energies (hereafter PE
and KE) of the water parcel have been interconverting with total energy dissipating continuously in the
tank. KE and PE of the entire tank are obtained from the following formulas.

$$KE = \int\limits_{0}^{L} \int\limits_{-H}^{\zeta} \frac{1}{2}\rho(u^2 + w^2)dxdz \qquad (27)$$

$$PE = \int\limits_{0}^{L} \int\limits_{-H}^{\zeta} \rho g z\, dxdz \qquad (28)$$

Above the formulas $\zeta$ is the free surface elevation. The three curves show the fluctuation of PE,
KE, and total energy during the K-H instability simulation in Fig. 3. The PE and KE correspond to the
maximum and zero due to the initial density distribution and static field in Fig. 3a. Afterward, the PE
declines sharply with an opposite change of KE. Both rates of change are almost the same based on the
curve slopes, which demonstrate that PE is converted to KE, reaching mutual peaks of about 9.5 s at the
end of the first gravitational adjustment. From then on, both of them still maintain the opposite trends
with an oscillation of roughly 25 s. In addition, it is worth noting that all kinds of energy show a
downward trend with their oscillation time increasing steadily due to the energy dissipations, which
seems that their time change rates overall diminish gradually so that KE will drop to zero and PE and
total energy (PE+KE) will reach the constant in the end. The results above are similar to the previous
works (Harel et al., 2000; Fringer et al., 2006; Lai et al., 2010), implying the correctness of the
nonhydrostatic dynamic module.



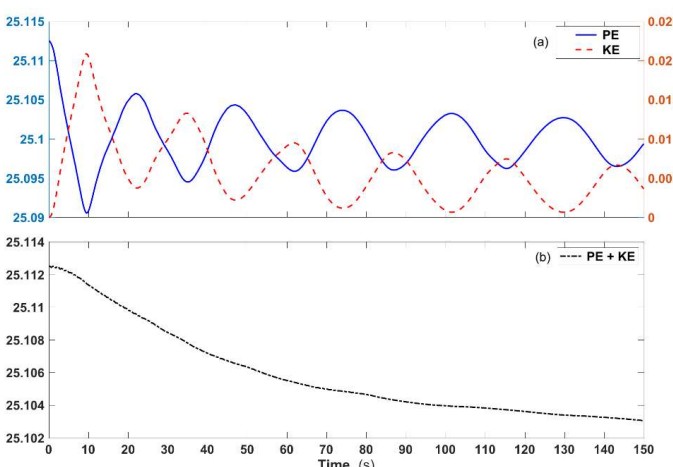


Figure 3. (a) The timeseries of the kinematical (red dashed line), potential (blue solid line) energy, and

(b) the same as total (black dotted line) energy (units: kg m$^2$ s$^{-2}$).

**3.2. Internal Solitary Wave in a tank**

Internal Solitary Waves are ubiquitous in the ocean with nonlinear and nonhydrostatic dynamic

processes. The laboratory experiments are usually carried out to study the ISWs to make up for the
defects of field observations. Additionally, the numerical study in a laboratory-scale experiment also
needs to be combined (Grue et al., 2000). We follow the previous experimental configuration (Ma et al.,
2019). The tank length is 2.0 m, and the static height is 10 cm without topographic change; The horizontal
and vertical resolutions are 2×10$^{-3}$ and 1×10$^{-3}$ m. Here, a gravity collapse method is used to generate the
ISW. Specifically, the low- and high-density fluids initially fill the upper and lower layers on the tank
with the collapse area on the left side. The collapse height and width are 5.0 cm and 4.0 cm, which will
lead to the generation of depression wave type. A schematic diagram of the ISW experiment is given in
Fig. 4. Water density averages are calculated in the upper and lower layer with $\rho_1 = 1003.62$ kg m$^{-3}$ and
$\rho_2 = 1026.95$ kg m$^{-3}$. Additionally, the diagnostic module is used to depict the high-frequency variation.
The high-frequency outputs are set at points $x$ = 0.4, 0.8, 1.2, and 1.6 m from west to east with an interval
of 0.05 s.

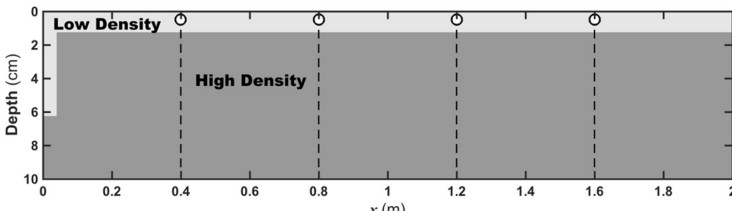


Figure 4. Schematic diagram of ISW case. The dark and light gray indicate the high- and low-density

water, where four white dots refer to the high-frequency output points.

Figure 5 clearly illustrates the evolution of the ISW packet in the tank based on the pycnocline

fluctuation. The isopycnic of 1026 kg m$^{-3}$ can represent the maximum strength of depression waves in
Fig. 5a. The depression wave packet from the west gravity collapse area comprises of the heading wave
and several tail waves with their amplitudes decreasing successively. The heading ISW with the
maximum amplitude propagates faster than the tails behind so that the distance increases between them.
As is exhibited in Table 1 about the heading depression wave characteristics at the four locations, we
find the amplitude of the depression wave with little change and then a slight increase but both no more
than 0.1 cm after $x = 0.8$ m, and the qualitative evaluation of the depression wave speed can be obtained
from the slope of the blue dashed area in Fig. 5b. Because the ISW packets are still at the stage of gravity
adjustment before arriving at $x = 0.2$ m, then wave speed increases slowly after $x = 0.2$ m but with its
increment less than 0.01 m s$^{-1}$. The above indicates that the internal solitary wave from gravity collapse
can propagate to the east steadily in our simulation. Besides, the characteristic westward reflected waves
(the blue line in Fig. 5a) with the larger amplitude prove that the wave-wave interactions happen between
the reflected and starting tail waves.

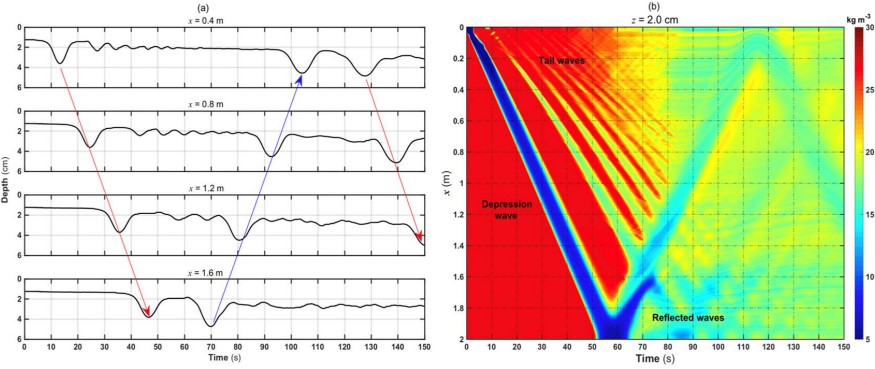


Figure 5. (a) The density timeseries of 1026 kg m$^{-3}$ at the four high-frequency output locations from the





west to east. The left red and blue arrow lines indicate the eastward and westward waves, and the right
red means the eastward reflected waves from the channel start. (b) Hovmöller diagram showing the
density at $z = 2.0$ cm where the time interval is 0.1 s.

Table 1 The characteristics of the depression heading wave at the four points

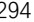

| location ($x$/cm) parameters | 0.4 | 0.8 | 1.2 | 1.6 |
|---|---|---|---|---|
| amplitude ($a$/cm) | 2.369 | 2.362 | 2.392 | 2.469 |
| characteristic wavelength ($L$/cm) | 19.632 | 21.643 | 23.206 | 25.822 |
| nonlinearity ($\varepsilon$) | 0.237 | 0.236 | 0.239 | 0.250 |
| dispersion ($\mu$) | 0.259 | 0.213 | 0.186 | 0.150 |


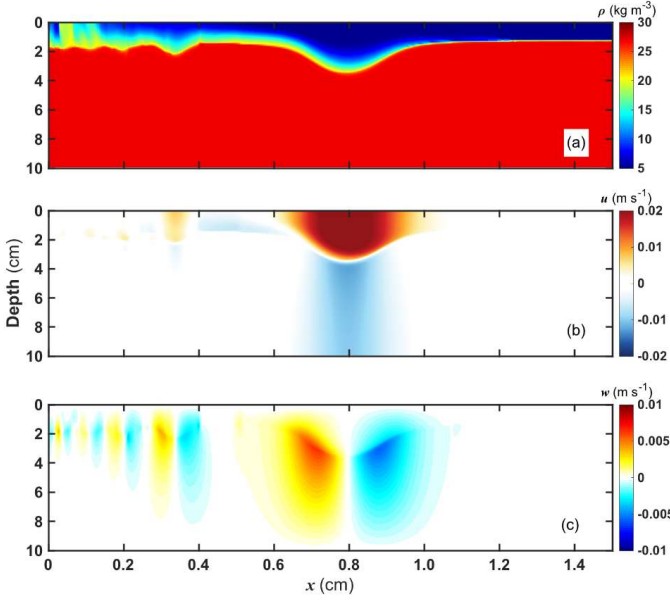


Figure 6. From top to bottom, density, horizontal and vertical velocity fields of the ISWs at $t = 24.5$ s.

We select a snapshot result for verification when the heading wave arrives at $x = 0.8$ m shown in
Fig. 6. The strongest horizontal velocity of the depression wave is 0.023 m s⁻¹, and the vertical flow can
reach up to 0.0065 m s⁻¹. The characteristic velocity fields are in line with the clockwise structure of a
theoretical depression internal solitary wave. Furthermore, the nonlinearity $\varepsilon = a/h$ and dispersion
$\mu = (h/\lambda)^2$ are calculated at the different locations in Table 1, where $a$, $h$, and $\lambda$ are the amplitude,





water height, and characteristic wavelength, and the KdV model (Benjamin, 1966) described in Appendix
B is used to calculate theoretical waveforms at the four locations depicted in Fig. 7. The comparison
demonstrates that the results are more consistent with the KdV model than m-KdV model. According to
the nonlinearity $\varepsilon$ from Michallet and Barthélemy (1998), the small and large-amplitude ISW can be
classified when $\varepsilon < 0.05$ and $\varepsilon > 0.05$, respectively. Whereas the application of the KdV model
requires a balance between the weak nonlinearities and dispersion (Ono, 1975), namely, assuming
that $\mu = O(\varepsilon) \ll 1$. Despite the large-amplitude waves simulated from our model with $\varepsilon > 0.05$, the
nonlinearity and dispersion are of the same order and small enough that the heading wave can be deemed
under weak nonlinearity, which is why their waveforms are better described well by the KdV model.
Therefore, the analysis results indicate that the simulated internal solitary wave is close to the KdV theory
and can be performed accurately via our nonhydrostatic model.

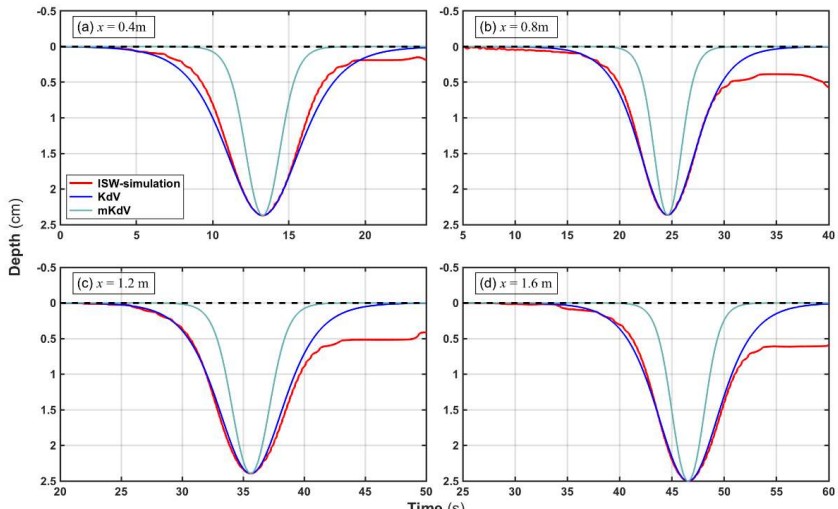


Figure 7. The interface displacement induced by ISW at four high-frequency output locations. The red
lines indicate the 1026 kg m$^{-3}$ isopycnic, and the blue and cyan lines represent the KdV and m-KdV

model results.

**3.3. Internal Solitary Wave shoaling on a Gaussian terrain**

Based on the experiment configuration in section 3.2 (also called Exp. 3.2). Here, a slowly varying

terrain is implemented to explore the nonlinear evolution of internal solitary wave, especially the wave
shoaling. As shown in Fig. 8, the left half of the Gaussian curve is reserved as the slope-shelf terrain



starting between $x = 1.0$ and 1.3 m with the height of 5.0 cm, and then the water depth remains unchanged
from $x = 1.3$ to 2.0 m corresponding to the shallow water zone. The high-frequency outputs are acquired
during the climbing process of ISWs at points $x = 0.4, 0.8, 1.0, 1.2, 1.3, 1.4, 1.6$, and 1.8 m from the west
to east with the output same interval as Exp. 3.2.

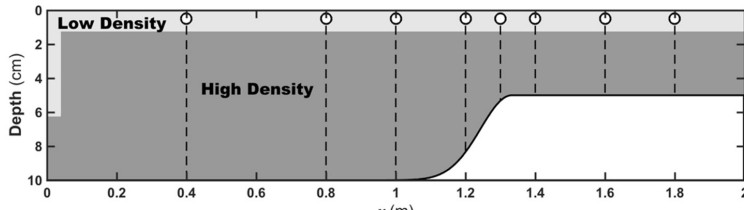


Figure 8. As in Figure 4, but with half-Gaussian topography in the east of the tank, where eight white

dots refer to the high-frequency output points.

Figure 9 reveals the evolution of the internal solitary waves with varying topography. The heading

ISW holds a stable packet at $x = 0.4$ m and begins to shoal after reaching the position of 1.0 m. Afterward,
its bottom velocity firstly experiences the topographic change, resulting in the different wave front and
rear effects. Specifically, the speed of wave trough is less than the rear, contributing to the wave front
gentle sloping but wave rear gradual steepening, which shows a similarity with Vlasenko et al. (2002).
Then the closed isopycnic contour mirrors the overturning and rolling due to the wave breaking in the
rear at $x = 1.2$ m in Fig. 9a. Apart from the overall wave breaking process above, it is also found in Fig.
9b that the reflected waves propagate to the west at $x = 1.2$ m due to the terrain shoaling. In other words,
the wave breaking and refection both lead to the depression wave energy being attenuated substantially.
When arriving at the east of $x = 1.2$ m, the original depression wave past the critical point so that an
elevation wave springs up in the wave rear, where the upper layer is thicker than the lower one in Fig.
10a. Then the elevation wave continues to propagate eastward. The high-density water accumulates the
upper water increasingly on the right wall, forming a collapse area between $x = 1.8$ m and the east wall,
where the thickness of the upper layer is larger than lower layer. Ultimately, the reflected waves including
a series of elevation tail waves, are released at $x = 1.6$ m. In detail, the first elevation is the leading one
with the rank-ordered structure in the rear all propagating to the west. After reaching the deep-water zone
left to 1.3 m, the wave rear begins to steepen and sink, generating a depression wave behind it. Namely,
the soliton wave passes the critical point inversely as a consequence of the water deepening.





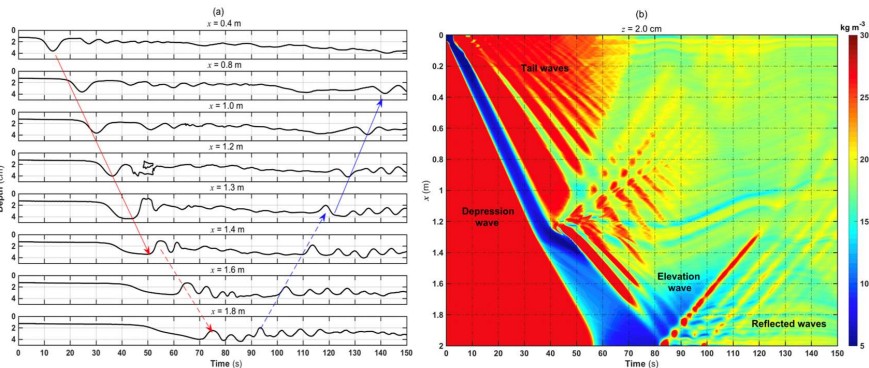

Figure 9. As in Figure 5, (a) the solid and dashed arrow lines indicate the depression and elevation waves, and the red and blue mean the starting westward and reflected eastward waves. (b) Hovmöller diagram showing the density at $z = 2.0$ cm.

For the sake of further exploration of the evolution of depression wave, the distributions of the vorticity ($\zeta = \partial w/\partial x - \partial u/\partial z$) with velocity vector are displayed in Fig.10. The soliton features a depression corresponding to negative vorticity with an anticyclonic structure before reaching the shelf topography. And then, the vertical shear increases rapidly and strengthens the positive vorticity at the bottom when the ISW closes to the top of the slope in Figs. 10a and 10b. As a consequence of shoaling, the backward overturning springs up between $x = 1.2$ and 1.3 m, marking the ISW entering the breaking instability stage (Helfrich and Melville, 1986). At this time, even though wave breaking and reflection cause the wave energy dissipation partially, the fraction of the depression wave can reach the shallow water zone, leaving behind a cyclonic vortex above the slope-shelf in Fig. 10c. This partial soliton wave is adjusted quickly when the thickness of the upper layer is more significant than the lower's in the light of the boundary of the negative vorticity area in Fig. 10d. As a result, the elevation wave begins to emerge at the back of the original wave, corresponding to the positive vorticity with a cyclonic structure when the reverse situation occurs. Notably, the vortex from the breaking of the depression wave weakens slowly and motivates the small-scale waves with high wavenumber propagating to both sides in Figs. 10e and 10f, which is consistent with the propagation characteristics of the reflected waves near $x = 1.2$ m in Fig. 9b.

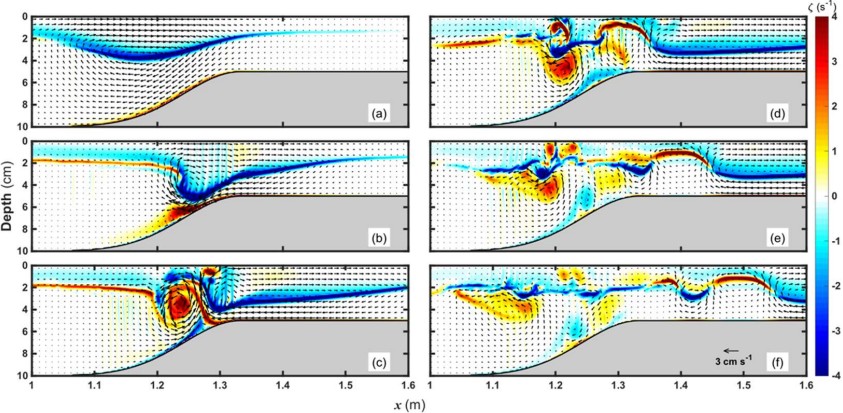

Figure 10. The shoaling of a depression soliton where the velocity fields (black arrow) and the vorticity results

(color) are shown at $t$ = (a) 35, (b) 40, (c) 45, (d) 50, (e) 55, and (f) 60 s.

It is also worthy of highlighting the evolution of the reflected eastward waves. We also visualize

the process of the second reverse situation due to the wave deepening in Fig. 11. It can be seen that there

is a strong elevation wave at $x$ = 1.4 m propagating to the deep-water zone with the crest corresponding

to positive vorticity, followed by a series of rank-order waves showing a sinusoidal variation. Particularly,

the wave train can be considered linear approximatively based on the alternated positive and negative

vorticity with the water interface almost located in the middle layer. The nonlinear parameter $\alpha$ in the

wave train region is close to zero in terms of the KdV model. As the depth becomes deeper, the crest of

the elevation wave gradually grows down and flattens with the wave rear sinking. The original elevation

cannot be maintained in the deep water, transforming into a depression wave with the velocity fields

adjusted accordingly.

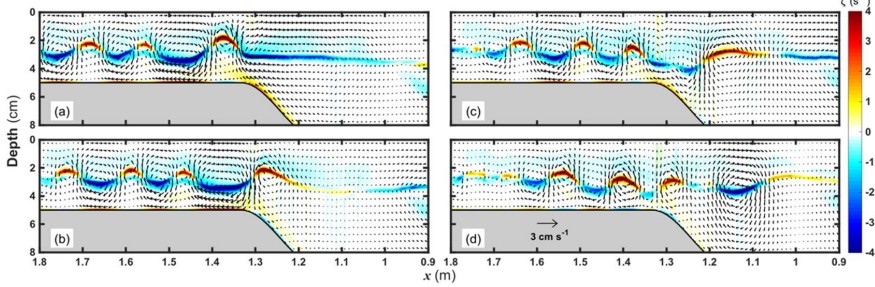

Figure 11. As in Figure 10, the elevation wave propagates westward to the deep water where the $x$-axis is inverse

for convenience at $t$ = (a) 115, (b) 120, (c) 125, and (d) 130 s.



Because the ISW in the adjustment stage is not stable enough to coincide with the KdV model after
pasting the critical point, we select the two types soliton results for verification before the reverse
situation occurs. The comparison between theoretical and numerical results is shown in Fig. 12 at $x = 0.8$
and 1.4 m before the wave shoaling and descending, respectively. We can find that the depression
waveform conforms to the KdV model results before climbing the slope, whereas the elevation is close
to the m-KdV model. Compared with the position of $x = 0.8$ m with $\varepsilon = 0.233$, a stronger nonlinearity
$\varepsilon = 0.331$ at $x = 1.4$ m in the shallow water could be the result of interaction between the ISW and the
varying topography. Actually, a larger ratio of amplitude in the shallow water proves results closer to the
m-KdV theory, which compares well with the conclusions of Michallet and Barthélemy (1998) in a
satisfactory way.

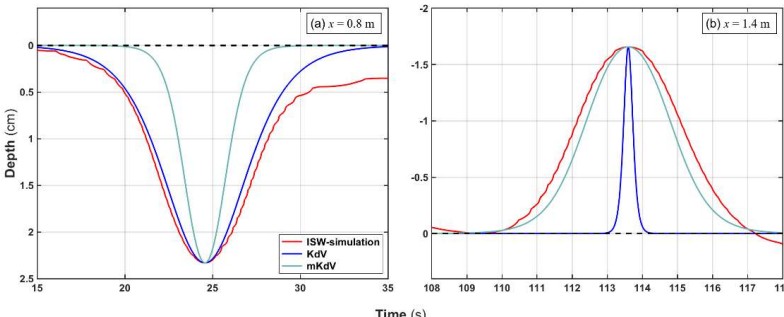


Figure 12. Wave profiles at $x = 0.8$ (a) and 1.4 m (b). The left refers to the depression heading wave
before shoaling and the right is the reflected elevation heading wave in the shallow water, which both
are plotted in red line, while and the blue and cyan lines represent the KdV and m-KdV model results.
**3.4. Internal Solitary Wave breaking on a slope.**
To further characterize a complete breaking and dissipative process of ISWs due to wave climbing,
we set a linear slope identical to Michallet and Ivey (1999). As is shown in Fig. 13, the tank length is 2.0
m; The height is 15 cm with the linear terrain placed on the east side. The depression wave is to be
dissipated due to increasing bottom friction at the shelf break. The spatial resolution is the same as Exp.
3.2, which can ensure the same timestep according to Courant-Friedrichs-Lewy (CFL) condition. In
addition, to compare with Bourgault and Kelley (2004) model results, water density averages are
calculated to be $\rho_1 = 1000.01$ kg m$^{-3}$ and $\rho_2 = 1047.00$ kg m$^{-3}$ in the upper and lower layers. Via several



sensitivity experiments about collapse area, the amplitude of depression wave can reach approximately
2.8 cm when the collapse height is 9.0 cm with its width same as Exp. 3.2. Although the stimulated wave
strength is slightly greater than the results from Bourgault and Kelley (2004) due to the different wave
generation methods, it is predictable that the breaking of the larger-amplitude ISW will be more dramatic
with a prominent performance for model verification.

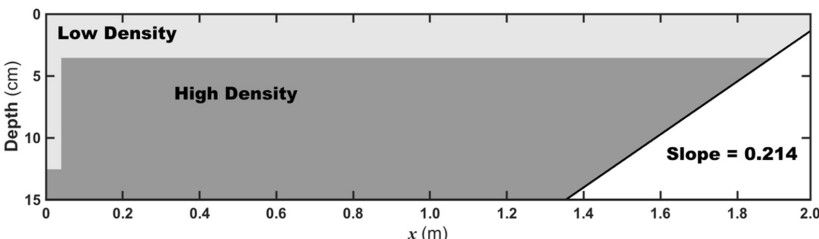


Figure 13. As in Figure 4, but with a linear slope terrain in the east of the tank and related configuration
is referred by Bourgault and Kelley (2004)
The corresponding density and velocity fields produced by the depression soliton are presented in
Fig. 14 at this moment before wave shoaling. The horizontal velocity induced is about 3.0 cm s$^{-1}$ at the
surface and varies up to 3.5 cm s$^{-1}$ at the core of the wave trough. Meanwhile, the vertical velocity
distribution presents a double-core structure reaching ±0.8 cm s$^{-1}$. The unique anticyclonic velocity
characteristic is entirely consistent with the experimental results of Bourgault and Kelley (2004). We
select the four different times with the evolution of wave shoaling illustrated in Fig. 15. In addition to
wave breaking accompanied by the waveform steepening in the rear, a significant density fronts rolling
in the wave front evolves along the linear slope during the overall shoaling process in Figs. 15a and 15b.
Specifically, while the depression wave continues getting closer to the shallow zone, the effect of bottom
friction can maintain a vertical shear with the potential energy increasing. Then the wave-induced
diapycnal flow contributes to high-density water under the interface transported continuously to the
shallow zone right to the slope in Fig. 15c, which intensifies the diapycnal mixing and dissipation on the
density interface. On the other hand, there is another pronounced change in Fig. 15d compared to the
experiment in Exp. 3.2. A few small-scale eddies emerge along with the sheared interface in density
fronts on account of the shear instability but are dissipated promptly via turbulent viscosity.



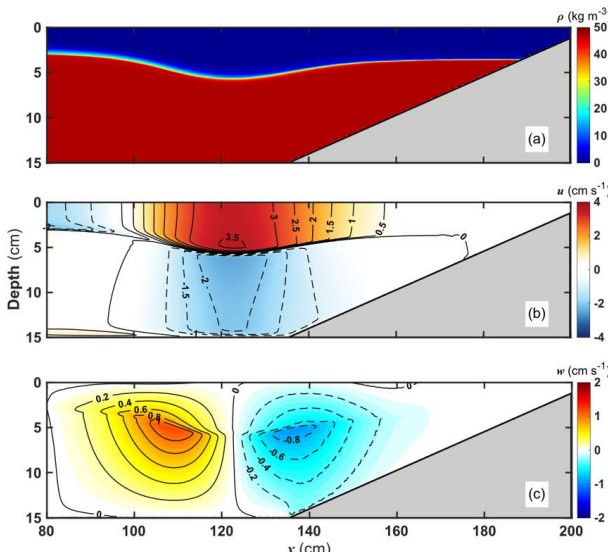


Figure 14. As in Figure 6, but with the time referring to $t$ = 15 s before the wave shoaling.

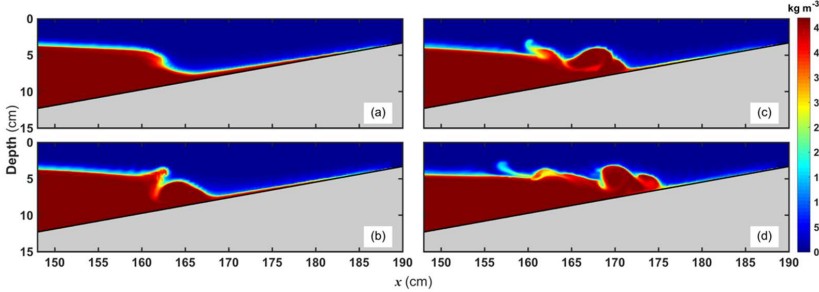


Figure 15. Wave breaking with density front rolling at $t$ = (a) 22, (b) 23, (c) 24, and (d) 25 s.
We compare the velocity field distributions with the observation results via PIV technology from
Michallet and Ivey (1999) and nonhydrostatic numerical experiments from Bourgault and Kelley (2004)
in Fig. 16 to further evaluate and validate the wave breaking process. Accordingly, when the
depression wave arrives over the slope, its depression waveform and anticyclonic flow field are
modulated by the topographic shoaling to flatten the wave front and enhance the current down the slope.
Meanwhile, the smaller cyclonic eddy appears and clings to the slope under the steepen wave rear.
Furthermore, as the depression wave continues shoaling, the cyclonic eddy strengthens and expands its
scope of influence, resulting in a strong overturning from near the bottom layer to promote the wave
steepening, which presents a good agreement with the results from Sect. 3.3. We also found that the
anticyclonic flow field persists in weakening as an effect of bottom friction. In contrast, the scope of the





cyclonic eddy expands and moves the shallow zone with the waveform distorted furtherly. In general, all
the above nonlinear processes are similar to the previous laboratory and model results. Our
nonhydrostatic model can also resolve the nonlinear evolution of the internal solitary waves at shelf break
with a high accuracy.

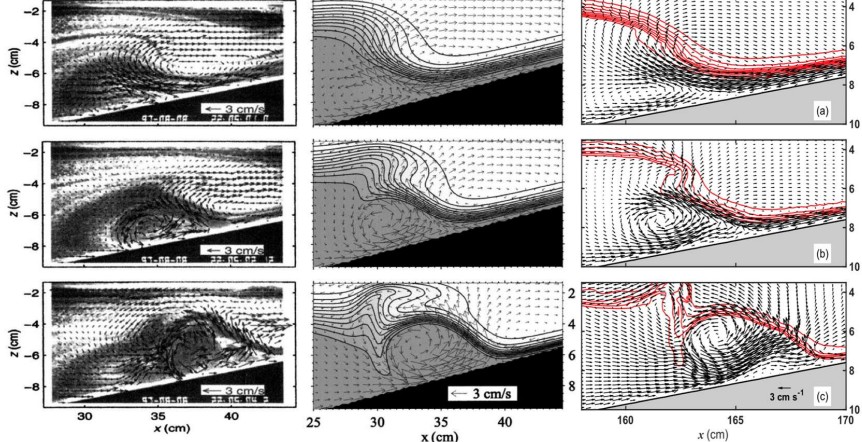


Figure 16. Comparison of velocity fields during the wave breaking on a linear slope between (left) the
PIV observations in the laboratory (Michallet and Ivey, 1999), (middle) the numerical model simulation
(Bourgault and Kelley, 2004), and (right) the ORCTM simulation at $t$ = (a) 21.7, (b) 22.2, and (c) 23.2 s

from top to bottom. The red contours indicate the isopycnic lines.




### 3.5. Nonlinear Internal Waves in a double-ridge system


The last validation experiment is to examine the generated nonlinear internal waves via tidal flow
over the varying topography. We set up an underwater double-ridge system comparable to the Luzon
Strait in the northern South China Sea. Meanwhile, this validation case is considered to be a 2-D problem
for the reduction of computational costs. The topography in this double-ridge system is fitted
approximately with the Gaussian function given as

$$H(x) = 3000 - h_w \times exp\left(-\left(\frac{x-x_w}{20 \times 10^3}\right)^2\right) - h_e \times exp\left(-\left(\frac{x-x_e}{20 \times 10^3}\right)^2\right) \qquad (29)$$

In Eq. (29), $H(x)$ is the water depth; the height of the East and West Ridge ($h_e$ and $h_w$) is 2500
and 1300 m in sequence with an interval and widths of 100 km, which is similar to the real topographic
characteristics in the Luzon Strait. As shown in Fig. 17a, the static water height is 3000 m, where the
East and West Ridge (hereafter ER and WR) are located at the coordinate origin and $x = $ -100 km; the
horizontal and vertical grid resolutions are 200 m and 10 m. Besides, to simplify the background
environment and get closer to the natural wave source fields, we use the horizontally uniform
stratification as the initial field to drive our model. Here, the reprehensive stratification in Figs. 17b to
17d stems from the summer stratification in 2011 of GLORYS12V1 reanalysis product in CMEMS
(Copernicus Marine Environment Monitoring Service) as the spatial mean around the source of generated
ISWs in the Luzon Strait, since the large-amplitude ISWs are observed during the same period on the
SCS continental shelf and the strong thermocline structure in summer is usually conducive to the
formation of baroclinic tides in the Luzon Strait (Zheng et al., 2007; Buijsman et al., 2010b; Ramp et al.,
2019). As for the tidal categories, owing to the generation of the semidiurnal internal tide with the
modulation of the diurnal one (Buijsman et al., 2010a; Zeng et al., 2019) in the Luzon Strait, we define
the M$_2$ and K$_1$ tidal currents amplitudes as 5.0 and 4.0 cm s$^{-1}$ corresponding to the semidiurnal and diurnal
components at the two open boundaries whose sponge thicknesses are both approximately 40 km.
Additionally, the slope criticality $\gamma$ (Gilbert and Garrett, 1989; Shaw et al., 2009) no less than one is
usually essential with the formation of linear internal waves.

$$\gamma = \frac{dH}{dx} \Big/ \sqrt{\frac{\omega^2 - f^2}{N^2 - \omega^2}} \qquad (30)$$

in which $\omega$ is the tidal angular frequency; $N^2$ is the buoyancy frequency squared; $f$ is zero without
regard to rotation. Around the East Ridge $\gamma$ is larger than unity regardless of the M$_2$ and K$_1$ tide with



the supercritical topography. Therefore, it is predictable to generate the internal waves due to the
interactions with barotropic flow over the East Ridge. We run the model for 10 days from initial static
fields. The diagnostic module is also used to depict the high-frequency variation with the output interval
of 1 min at $x$ = -250, -350 km.

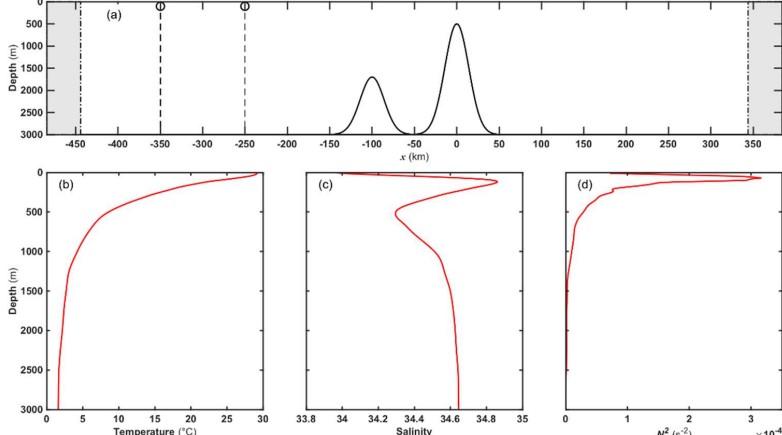


Figure 17. (a) The sketch of generated NIWs over the submerged double-ridge system case, and the
gray zones indicate the sponger layers. The summer stratification in 2011 including (b) temperature, (c)
salinity, and (d) buoyancy frequency squared are from the spatial mean within 20.25 °N–20.85 °N,
121.7 °E–122.08 °E corresponding to the source of internal waves in the Luzon Strait (Zhang et al.,

2011).


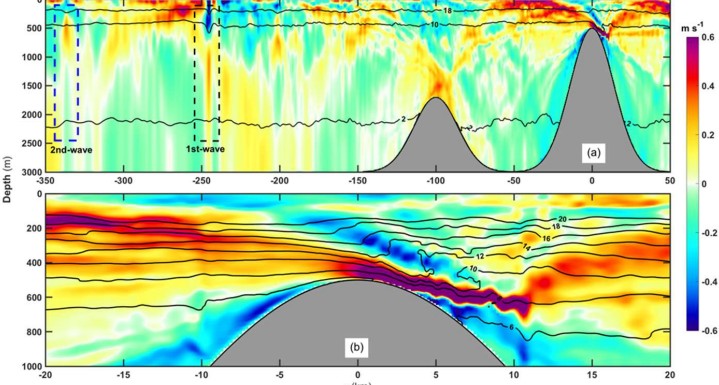


Figure 18. Distributions of horizontal baroclinic velocity with the temperature (°C) contours for
western far-field (a) and source field (b) when the maximum eastward tidal current at East Ridge

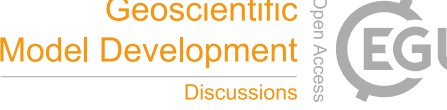

reaches the end of ebb on the sixth day, where the blue (black) dashed box means the 2nd mode (1st

mode) ISW packets.

Figure 18 shows the maps of horizontal baroclinic velocity $u' = u - U$ where $u$ is the total

velocity and $U$ is the barotropic flow velocity. From the characteristics of the source field, it is found
that the generation of internal tide beam propagating eastward and westward centered from the eastern
side of East Ridge. The eastward barotropic flooding current flows continuously over the east ridge with
the maximum barotropic current up to 0.0531 m s$^{-1}$. The hydraulic jump can appear with the isotherms
fluctuation up to roughly 200 m on the eastern side, which indicates the formation of Lee waves to a
certain extent. Above internal waves generation due to tide-topography interactions can be described
with several non-dimensional parameters at the source: (1) the tidal excursion parameter $\varepsilon = U_0/L\omega$,
which can be associated with the generation of internal beam under the critical or super-critical
topography where $U_0$ is barotropic current amplitude from the far-field and $L$ is the characteristic
length for topography (Garret and Kunze, 2007, Chen et al., 2017). (2) the Froude number $Fr = U/c$,
and the topographic form $Fr_z = \omega/N(dH/dx)$, in which $c$ is the mode-1 linear speed for the
eigenvalue problem (Legg and Adcroft, 2003; see Appendix B). Specifically, Legg and Klymak (2008)
found that the nonlinear hydraulic jump will develop with lee waves generation when $Fr_z < 1/3$.
Moreover, it is worth paying attention to the tidal excursion far less than unity that agrees with the
formation of the linear internal beam on the critical or super-critical topography but cannot ensure the
appearance of the Lee waves altogether. For instance, the Lee waves remain strong in the Luzon Strait
despite the tidal excursion under the unity ($\varepsilon \approx 0.4$) in previous model results (Buijsman et al., 2010b).
the tidal excursion parameter $\varepsilon$ and the Froude number $Fr$ in our model are calculated to be 0.025 and
0.018, indicating the generation of multi-modal baroclinic tides and upstream propagation of internal
waves with the sub-critical barotropic current over the east ridge. Furthermore, the maximum
topographic Froude number is just 0.3362 around the East ridge with the approach to the regime transition
value 1/3, which ensures that the nonlinear hydraulic jump can develop with Lee waves on the east of
East ridge, which helps to explain well the evolutions of the internal beam and hydraulic jump in our
simulation and confirms to the mixed tidal lee wave regime in the Luzon Strait (Chen et al., 2017).



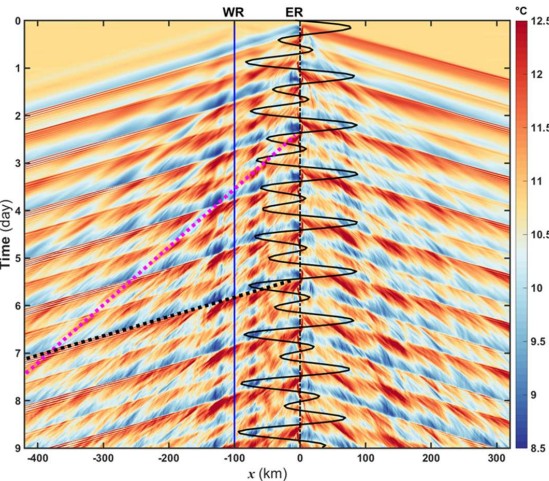


Figure 19. Hovmöller diagram about the global temperature timeseries at $z = 400$ m, where the time
interval is 15 mins. The black solid curve indicates the tidal current at the East Ridge, and the blue
solid line means the west ridge location. The black and magenta dashed lines are the first and second-
mode Internal Solitary Waves.
The westward internal tide beam emitted from the East Ridge reaches the sea surface and reflects
back into the deep sea, and the internal tide beam can propagate to the top of the West ridge below 1500
m depth and reflect back to the upper layer again. Between the double ridges such a more significant
portion of beam energy captured by the waveguide in the pycnocline can strengthen the upstream
influence to propagate horizontally westward in Fig. 18b, which can trace back to the source of the
internal solitary wave packets of the far-field. However, the strong dissipation for the high modal internal
waves contributes to the vanishing of the internal beam structure and allows the nonlinear evolution of
low-mode baroclinic tides. The significant internal solitary wave packets can appear and propagate
westward from -150 km, marking the disintegration of the nonlinear internal waves energy. Specifically,
the first-mode ISW packet emerges from $x = $ -250 to -200 km. Meanwhile, the second-mode ISW
between $x = $ -350 and -300 km performs the convex wave packet. We can achieve the propagation
characteristics of these ISWs via analyzing the global temperature timeseries at 400 m water depth layer
illustrated in Fig. 19. The second-mode ISW propagates slower and weaker than the first-mode wave
packet. Besides, it can be distinguished that the two first-mode wave packets can propagate westward,
one of which is stronger with the structure of several tail waves, and the other is almost solitary and weak,
symbolizing the observed changes in the structure and timing of type-a and b waves (hereafter a-wave



and b-wave) originating from the Luzon Strait (Ramp et al., 2004; 2019; Zhao and Alford, 2006). The
relatively weak second-mode concave wave can be found distinctly following the a-wave from the west
of -300 km. To sum up, our multi-modal baroclinic tide structures originating from the double-ridge
system can propagate to the far-field and the low-mode internal waves gradually perform the
corresponding ISWs due to the nonlinear enhancement, which shows a good agreement with the other
two-dimensional experimental results (Buijsman et al., 2010a; 2010b; Vlasenko et al., 2010).

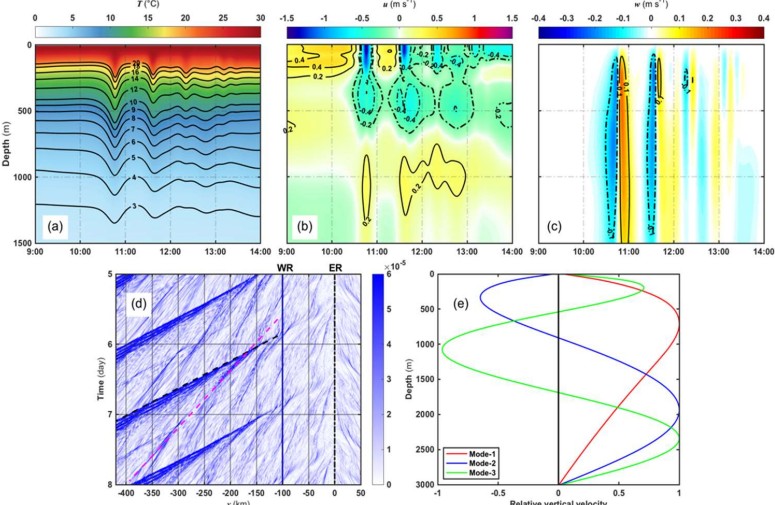


Figure 20. (a) The temperature (°C), (b) horizontal baroclinic velocity (m s$^{-1}$), (c), and vertical velocity

(m s$^{-1}$) structures of the first-mode ISW packet at $x$ = -250 km on the sixth day. (d) The SSHG

Hovmöller diagram during the corresponding period where the black and magenta dashed lines indicate

the first and second-mode ISW packets. (e) The normal mode profiles of vertical velocity for the first

three modes via the Taylor-Goldstein equation.

To evaluate the comparison between the numerical ISWs with internal wave theory, we choose the

results of the first-mode ISW at  $x$ = -250 km shown in Fig. 20. It is found that a first-mode ISW packet
including three tail waves arrives at the position after 10 a.m. on the 6th day. The maximum fluctuation
of the first-mode ISW packet can reach 206 m between 650 and 900 m water depths. The westward
horizontal baroclinic velocity associated with the wave packet prevails above 200 m with the maximum
strength of roughly 1.41 m s$^{-1}$, and the corresponding downwelling region is located between 200 and
1500 m with the strongest downward velocity up to 0.22 m s$^{-1}$. According to the Sea Surface Height
Gradient (SSHG, SSHG is defined $\sqrt{(\nabla\zeta)^2}$), the average propagation speed of this wave packet is



approximately 3.17 m s⁻¹ based on the slope of SSHG contour. Moreover, we solved the Taylor-Goldstein
equation (Miles, 1961; Liu, 2010; see Appendix B) at 10 minutes before this wave packet reached $x$ =
-250 km. The normal mode of vertical velocity is subject to the rigid-lib boundary condition illustrated
in Fig. 20e. it is found that the location corresponding to the maximum modal function is 710 m in
agreement with the model results. However, the propagation speed is greater than the first-mode linear
result of 2.69 m s⁻¹, which might be probably attributed to the linear theory with the underestimated effect.
Therefore, The KdV model is also used to analyze the depression wave, which shows that the dispersion
parameter $\beta$ is 2.4×10⁵ m³ s⁻¹, and the nonlinear parameter $\alpha$ = -3.4×10⁻³ s⁻¹ denotes that the predicted
depression wave is consistent with the simulated results (Helfrich and Melville, 1986). Nevertheless, the
theoretical nonlinear velocity of about 2.88 m s⁻¹ is slightly lower than the realistic propagation velocity
obtained from the simulated results. It is probable that the increasing nonlinearity with the steepening of
internal tides ultimately leads to the larger propagation speed of this first-mode ISW packet.

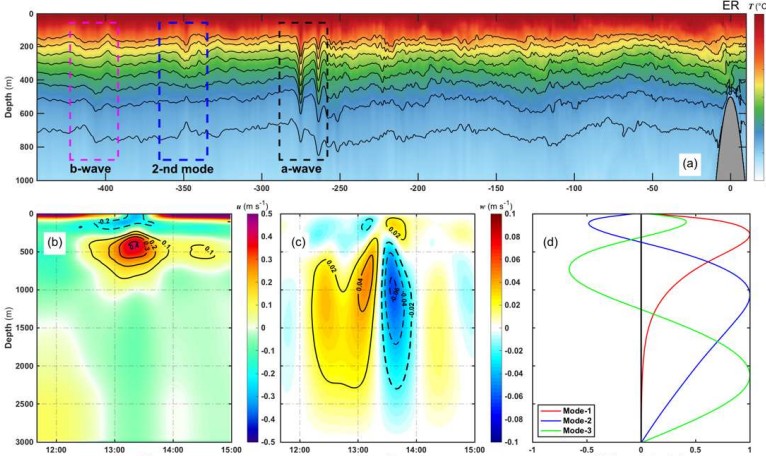


Figure 21. (a) The temperature (°C) field from the west side of East Ridge at 13:00 on the seventh day,
where dashed rectangles refer to the respective type waves. (b) The horizontal baroclinic velocity (m s⁻
¹) and (c) vertical velocity (m s⁻¹) structures of the second-mode ISW at $x$ = -350 km in the meantime.
(d) The normal mode profiles about vertical velocity for the first three modes via the Taylor-Goldstein

equation.

It is also noticeable that the multi-modal structure of baroclinic tides generated from the double-

ridge system can evolve into the high-mode internal solitary waves due to the nonlinear enhancement.
We can recognize clearly the distinct ISW packets from the isotherm displacement that refers to the type-





a, second-mode, and type-b waves from the source to the far-field in Fig. 21a. The a-wave packet features
the most substantial strength with tail waves when its vertical excursion induced by the heading wave
can reach up to 120 m. In contrast, the weaker b-wave contains one solitary depression wave in the west
to $x$ = -400 km. They both originate from multi-modal internal tide caused by the tide-topography
interactions in the double-ridge system, but the b-wave is more associated with the west ridge (Buijsman
et al., 2010a; Zeng et al., 2019). Between a- and b-wave, there is a second-mode ISW packet classified
obviously as a structure of concave wave whose upper and lower isotherm is toward downward and
upward. The isotherm fluctuation can reach up to -57.2 and 140.6 m in the upper and lower layers
referring to the rough depths of 180 and 1000 m. Based on the slope in Fig. 21d, the propagation speed
of this second-mode ISW signal is about 1.36 m s$^{-1}$. It is predictable that the a-wave packet will follow
the second-mode signal due to the larger speed. Namely, the first and second mode solitary waves as the
leading carriers transferring energy from the source to far-fields until dissipating thoroughly. The multi-
modal solitary waves field agrees with the previous two-ridge experimental results via the MITgcm
(Vlasenko et al., 2010), which indicates that the multi-modal baroclinic tides propagate westward with
the low-mode signals evolving into the ISWs.
The second-mode ISW packet with corresponding velocity fields timeseries at $x$ = -350 km are
shown in Figs. 21b and 21c. The horizontal baroclinic velocity field has a sandwich-shaped vertical
structure, and the maximum 0.42 m s$^{-1}$ is located in the middle layer between 200 and 600 m.
Nevertheless, the baroclinic velocity above 200 m is distinct from the first-mode ISW packet with a small
value of 0.2 m s$^{-1}$. Additionally, a double-peak structure performs and is distributed at the depths of 150
and 1000 m from the vertical velocity field where the strength in the deep layer is stronger than the
upper's, resulting in the smaller isotherm fluctuation above 200 m. Here, the Taylor-Goldstein equation
is also used to acquire the eigenfunction about the vertical velocity shown in Fig. 21d. The second-mode
eigenvalues have two peaks in vertical direction whose depths correspond to 150 and 1070 m with the
latter strength stronger than the former, and the corresponding phase speed is about 1.34 m s$^{-1}$. The above
theoretical results can compare well with the distribution of stimulated characteristics, indicating that our
nonhydrostatic ocean model succeeds in describing the nonlinear evolution of higher modal baroclinic
tides.





### 4. Conclusion


In this paper, a reliable ocean model called ORCTM which is able to reproduce nonhydrostatic
dynamics with the marched open boundary condition to depict the evolution of the internal solitary waves,
is introduced. Based on the fractional step method, the three-dimensional fully nonlinear momentum
equations are involved and considered thoroughly under the Boussinesq fluid. It is needed to solve the
three-dimensional Poisson equation subject to different boundary conditions before the pressure
correction method is employed to acquire the velocity field corrected via nonhydrostatic pressure
gradients at the new timestep. In order to match the nonhydrostatic algorithm and realize larger-amplitude
ISWs simulation in an ocean-scale case, an exponential relaxation term is implemented to the control
equations through the sponge layers as the open boundary condition.
A series of two-dimensional ideal numerical experiments corresponding to the nonlinear evolution
of the internal solitary waves and baroclinic tides are designed to verify the nonhydrostatic module. Here,
the results of the validation experiments can be in accord with the theoretical framework of the
nonhydrostatic dynamics and demonstrate that the ORCTM can successfully reproduce the generation,
propagation, and dissipation of Internal Solitary Waves in laboratory-scale cases. Specifically, the reverse
situation due to wave shoaling and deepening can be compared well to the previous simulation when
considering the topographic change. Meanwhile, the ORCTM can capture the density fronts with the
cyclonic eddy induced by the wave breaking, which shows enough accuracy with a good stability.
Notably, the multi-modal structure of baroclinic tides stems from the tide-topography interactions
in the double-ridge system based on the real topographic features in the Luzon Strait of the northern
South China Sea, where the largest internal solitary waves in the world can exist. This application of this
case suggests that our model can reproduce the life cycle of ISWs induced from the Luzon Strait and
capture the whole alternation process of type-a and b internal solitary wave packets. The first two mode
ISWs structure compares well to those derived from the internal wave theoretical model.
Compared to the in-situ observations, the simulation of internal waves can mirror the macroscopic
structure and assist with the implementation of observations. However, the predictability of nonlinear
internal waves characteristics relies on the model performance and reliable external conditions such as,
the realistic stratification, bathymetry, and background circulation. Enhancing the fidelity of ISWs
remains to be challengeable. Nevertheless, it can be concluded that this regional nonhydrostatic ocean





model can provide a choice for the oceanography scientists who are interested in internal waves research
and internal waves numerically prediction.

**Appendix A**
**Discretization Algorithms of the Poisson Equation**
According to the idea of fractional steps (Chorin,1968; Press et al., 1988), a pressure correct method
considering the nonhydrostatic component is employed to calculate the intermediate velocity over the
original hydrostatic balance scheme (Fringer et al., 2006; Lai et al., 2010). If the flow is close to the
hydrostatic balance, the pressure of nonhydrostatic part will be so slight that the correction plays a minor
role. The key to the nonhydrostatic dynamics module is to solve the Poisson equation below.

$$\frac{\partial^2 p'_{nh}}{\partial x^2} + \frac{\partial^2 p'_{nh}}{\partial y^2} + \frac{\partial^2 p'_{nh}}{\partial z^2} = \frac{\rho_c}{\Delta t}\left(\frac{\partial \tilde{u}}{\partial x} + \frac{\partial \tilde{v}}{\partial y} + \frac{\partial \tilde{w}}{\partial z}\right) \tag{A.1}$$

The right-hand side (RHS) of this Eq. (A.1) is the divergence about the intermediate velocity as a
source or sink term. Here, based on the definition about divergence, the three components calculated
directly at each cell are specified in the three orthogonal coordinates as

$$\frac{\partial \tilde{u}}{\partial x} = \frac{\tilde{u}_{i,j}^k * Au_{i,j}^k - \tilde{u}_{i-1,j}^k * Au_{i-1,j}^k}{\Omega_{i,j}^k} \tag{A.2}$$

$$\frac{\partial \tilde{v}}{\partial y} = \frac{\tilde{v}_{i,j-1}^k * Av_{i,j-1}^k - \tilde{v}_{i,j}^k * Av_{i,j}^k}{\Omega_{i,j}^k} \tag{A.3}$$

$$\frac{\partial \tilde{w}}{\partial z} = \frac{\tilde{w}_{i,j}^k * Aw_{i,j} - \tilde{w}_{i,j}^{k+1} * Aw_{i,j}}{\Omega_{i,j}^k} \tag{A.4}$$

In Eqs. (A.2) to (A.4), $i$, $j$ and $k$ are the indices of increasing eastward, northward, and
downward along $x$, $y$, and $z$-axis, respectively, where $z = 0$ is defined on the undisturbed sea surface by
means of Cartesian coordinate system. $\tilde{u}$, $\tilde{v}$, and $\tilde{w}$ are the intermediate velocity; $Au$, $Av$ and $Aw$
means the six faces area of a cell in $i$, $j$ and $k$ directions; $\Omega$ is the volume of a cell. Compared to the
finite difference method, the definition of the divergence of a cell is more accurate and reliable especially
when adjacent to the solid boundaries for the RHS calculation. The left-hand side (LHS) of this equation
is discretized horizontally on the Arakawa C-grid (Arakawa and Lamb, 1977) using the central difference
method with a second-order accuracy, and the vertical discretization is the same as Max-Planck-Institute
ocean model (Marsland et al., 2003), which is able to acquire the following finite discrete equation about
7 cells for nonhydrostatic pressure perturbation as





$$LHS = (XW)p'^k_{i-1,j} + (XE)p'^k_{i+1,j} + (YN)p'^k_{i,j+1} + (YS)p'^k_{i,j-1} + (ZU)p'^{k-1}_{i,j}$$
$$+ (ZD)p'^{k+1}_{i,j} + (XC + YC + ZC)p'^k_{i,j}$$

(A.5)

663  In Eq. (A.5) the coefficients of the discretized LHS are given as follows.

$$XW = \frac{1}{DXu_{i-1,j} * DXp_{i,j}}, \qquad XE = \frac{1}{DXu_{i,j} * DXp_{i,j}},$$

$$YN = \frac{1}{DYv_{i,j-1} * DYp_{i,j}}, \qquad YS = \frac{1}{DYv_{i,j} * DYp_{i,j}},$$

$$ZU = \frac{1}{DZw^k_{i,j} * DZp^k_{i,j}}, \qquad ZD = \frac{1}{DZw^{k+1}_{i,j} * DZp^k_{i,j}},$$

$$XC = -\left(\frac{1}{DXu_{i-1,j}} + \frac{1}{DXu_{i,j}}\right)\frac{1}{DXp_{i,j}},$$

$$YC = -\left(\frac{1}{DYv_{i,j-1}} + \frac{1}{DYv_{i,j}}\right)\frac{1}{DYp_{i,j}},$$

$$ZC = -\left(\frac{1}{DZw^{k+1}_{i,j}} + \frac{1}{DZw^k_{i,j}}\right)\frac{1}{DZp^k_{i,j}}$$

(A.6)

664  The $DX$, $DY$ and $DZ$ represent the spacing difference between the adjacent grid cells in $x$, $y$, and

665 $z$-axis. The suffixes associate $u$, $v$ and $w$ at cell face center and $p'$ at body center. Thus, invoking the

666 boundary conditions (20) and Eqs. (A.5) to (A.6), the discretized Poisson equation with 7 cells can be

667 derived with the matrix form below

$$Ap'_{nh} = B$$

(A.7)

668  Where $A$ is a sparse, and definite-positive matrix with seven diagonals; $p'_{nh}$ and $B$ are the

669 column vectors with a size of all cell number $Nxyz = Nx \times Ny \times Nz$ in the model domain where $Nx$,

670 $Ny$, and $Nz$ are the cell number in $i$, $j$ and $k$ directions. Actually, the sparse matrix $A$ is too huge to

671 handle directly with a size of $Nxyz \times Nxyz$, which needs to be designed with greater efficiency as a

672 precondition. To apply the nonhydrostatic model to the real oceanic environment on the original model

673 base, the Portable, Extensible Toolkit for Scientific Computation (PETSc) Library is implemented into

674 the nonhydrostatic dynamic module. We apply the numerical Krylov subspace methods for the matrix

675 solvers under an MPI-based framework (Balay et al., 2020). Here, the Flexible Generalized Minimal

676 Residual (FGMRES) method (Saad, 1993) is employed to solve the equation (A.7) in conjunction with

677 a multigrid preconditioner (Smith et al. 1996) for the sparse matrix before iteration. With these methods

678 the nonhydrostatic dynamics can be fulfilled economically in harmony with the original numerical

679 framework.



**Appendix B**

**The Korteweg–de Vries (KdV) Model in the Shallow Water**

Based on the shallow water approximation, a small-amplitude internal solitary wave whose amplitude compared with the total depth is small enough can be described by the classical two-dimensional Korteweg-de Vries (KdV) equation given as follows (Apel et al., 2007)

$$\frac{\partial \eta}{\partial t} + c\frac{\partial \eta}{\partial x} + \alpha\eta\frac{\partial \eta}{\partial x} + \beta\frac{\partial^3 \eta}{\partial x^3} = 0 \tag{B.1}$$

Considering two-fluid stratification system is more appropriate for the experiments in Sec. 3.1–3.3. $\rho_1$ and $\rho_2$ are the upper and lower densities corresponding to the thickness $h_1$ and $h_2$; $x$ is the horizontal coordinate. Several parameters can be written here as (Benjamin, 1966; Wessels and Hutter, 1996)

$$\alpha = -\frac{3c}{2}\frac{\rho_1 h_2^2 - \rho_2 h_1^2}{\rho_1 h_1 h_2^2 + \rho_2 h_1^2 h_2}, \beta = \frac{c}{6}\frac{\rho_1 h_1^2 h_2 + \rho_2 h_1 h_2^2}{\rho_1 h_2 + \rho_2 h_1}, c = \sqrt{\frac{gh_1 h_2(\rho_2 - \rho_1)}{\rho_1 h_2 + \rho_2 h_1}} \tag{B.2}$$

where nonlinear and dispersion parameters ($\alpha$ and $\beta$ respectively) can represent the soliton polarity; $c$ is the linear velocity and the solution of solitary wave is expressed below the interface displacement $\eta(x,t)$

$$\eta(x,t) = \eta_0 sech^2\left(\frac{x - Vt}{L}\right) \tag{B.3}$$

in which the $\eta_0$ is the amplitude. The nonlinear velocity $V$ (also called phase velocity) and the characteristic length of soliton $L$ are given as

$$V = c + \frac{\alpha}{3}\eta_0, \qquad L = \sqrt{\frac{12\beta}{\alpha\eta_0}} \tag{B.4}$$

The dispersion parameter $\beta$ is almost larger than zero for the internal solitary waves in the ocean but the sign for the nonlinear parameter $\alpha$ is relevant to the wave formation. When $\alpha > 0$, the interface displacement will show a waveform of depression soliton. If negative, the isopycnal elevation will appear. Therefore, the reverse situation for an internal solitary wave is determined by the sign change of the nonlinear parameter. The KdV model is suitable with weakly nonlinear and dispersive waves which is capable of being used to validate the small-amplitude ISW results in the laboratory. Nevertheless, when nonlinearity enhancement happens by the reason of shallower topography or stronger stratification, the modified KdV (m-KdV) model (Michallet and Barthelemy, 1998; GrimShaw et al., 2004) can describe



relatively stronger nonlinear solitons with the addition for cubic nonlinearity term as

$$\frac{\partial \eta}{\partial t} + (c + \alpha\eta - \alpha_1\eta^2)\frac{\partial \eta}{\partial x} + \beta\frac{\partial^3 \eta}{\partial x^3} = 0 \tag{B.5}$$

It is worthy of noting that the m-KdV equation takes the higher-order nonlinear term into account
and can degenerate into the KdV equation when the cubic nonlinear parameter $\alpha_1 = 0$. Here, the solution
is given with the interface displacement $\eta(x,t)$

$$\eta(x,t) = \frac{\eta_0 sech^2\left(\frac{x-Vt}{L}\right)}{1 - \mu\,tanh^2\left(\frac{x-Vt}{L}\right)} \tag{B.6}$$

where

$$h_c = \frac{h_1 + h_2}{1 + \sqrt{\rho_1/\rho_2}}, \qquad \bar{h} = h_2 - h_c,$$

$$\mu = \begin{cases} h''/h', & \bar{h} > 0 \\ h'/h'', & \bar{h} < 0 \end{cases},$$

$$h' = -\bar{h} - |\bar{h} + \eta_0|, \qquad h'' = -\bar{h} + |\bar{h} + \eta_0|,$$

$$V = c_{0m}\left[1 - \frac{1}{2}\left(\frac{\bar{h} + \eta_0}{h_1 + h_2 - h_c}\right)^2\right],$$

$$c_{0m} = \left\{\frac{g(h_1 + h_2)}{2}\left[1 - \left(1 - \frac{4h_c(h_1 + h_2 - h_c)(\rho_2 - \rho_1)}{\rho_2(h_1 + h_2)^2}\right)^{1/2}\right]\right\}^{1/2},$$

$$L = 2(h_1 + h_2 - h_c)\sqrt{\frac{(h_1 + h_2 - h_c)^3 + h_c^3}{3(h_1 + h_2)h'h''}} \tag{B.7}$$

More generally, when considering the continuously stratified fluid, the linear velocity $c$ refers to
the long-wave velocity of each mode for the Strum-Liouville problem given as follows (Apel et al., 2007)

$$\begin{cases} \dfrac{d^2W}{dz^2} + \dfrac{N^2}{c^2}W = 0 \\ W = 0, & z = 0 \\ W = 0, & z = -H \end{cases} \tag{B.8}$$

where $H$ is the water depth; $N$ is the buoyancy frequency; $W$ is the nondimensional modal function.
When the nonlinear and dispersion parameters ($\alpha$ and $\beta$ respectively) are obtained as

$$\alpha = \frac{3c\int_{-H}^0 (dW/dz)^3\,dz}{2\int_{-H}^0 (dW/dz)^2 dz}, \qquad \beta = \frac{c\int_{-H}^0 W^2\,dz}{2\int_{-H}^0 (dW/dz)^2 dz} \tag{B.9}$$

Besides, if still considering the background current $\bar{U}(z)$, the Taylor-Goldstein equation (Miles,
1961; Liu, 2010) can describe the vertical modal function $W$, when the nonlinear and dispersion
parameters are obtained under the Boussinesq approximation expressed as (GrimShaw et al., 2002).





$$\frac{d^2\hat{\varphi}(z)}{dz^2} + \left[\frac{N^2}{(\overline{U}-c)^2} - \frac{\overline{U}''}{(\overline{U}-c)} - k^2\right]\hat{\varphi}(z) = 0 \tag{B.10}$$

$$\alpha = \frac{3\int_{-H}^{0}(c-\overline{U})^2\left(\frac{dW}{dz}\right)^3 dz}{2\int_{-H}^{0}(c-\overline{U})\left(\frac{dW}{dz}\right)^2 dz}, \quad \beta = \frac{\int_{-H}^{0}(c-\overline{U})^2 W^2 dz}{2\int_{-H}^{0}(c-\overline{U})\left(\frac{dW}{dz}\right)^2 dz} \tag{B.11}$$

where $c$ is the n-mode linear speed; $\hat{\varphi}(z)$ is the stream function; $\overline{U}''$ are the second derivative of
background currents; $k$ is the horizontal wave number.



**Code and data availability.**


The current version of the nonhydrostatic ocean model (ORCTM-v1) and these experiments about
the internal solitary wave simulation in this paper are available through
https://doi.org/10.5281/zenodo.6683597 (HaoHuang, 2022), as well as the experiment configurations,
preprocessing, and post-processing. The PETSc library (the download address:
https://petsc.org/release/download/, Balay et al., 2020) needs to be installed before building the model.
Nevertheless, we also provide the PETSc library of the version in use and the ORCTM quick manual for
the users at the above link.

**Author contributions.**


HH and PS developed the nonhydrostatic dynamic framework in ORCTM and devised the internal
solitary wave validation experiments. SQ and JG developed the open boundary module. HH and SQ
analyzed the model results and interpreted the concepts, and all authors contributed to the writing of the
paper.

**Competing interest.**


The authors of this paper declare that they have no conflicts of interest.

**Acknowledgements.**


This study has been conducted using the E.U. Copernicus Marine Service Information global ocean.
Specifically, the GLORYS12V1 product in CMEMS eddy-resolving reanalysis is extracted during the
summer of 2011 whose DOI is: https://doi.org/10.48670/moi-00021.We also thank both National
Supercomputing Center in Jinan and the Marine Big Data Center of Institute for Advanced Ocean Study
at Ocean University of China for the provision of computing resources.

**Financial support.**


This research has been supported by the National Key Research and Development Program of China,
Grant 2021YFF0704002 (Super-resolution assimilation and fusion model for ocean data, SAFMOD) and
the National Natural Science Foundation of China (NSFC), Grant No. 9195820006.

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
