# Peer review of "A nonhydrostatic oceanic regional model ORCTM v1 for"

_Geoscientific Model Development, 2022_

## Author Response (AR1)

**Response letter to the referee #1**

*This paper presents a nonhydrostatic oceanic model for internal waves based on the framework of the Max-Planck-Institute ocean model. I suggest that the paper can be accepted after minor revision.*

We would like to thank the referee for this careful reading and valuable comments. In the revision, we have added more details on the numerical test cases to clarify the validation experiment configurations. We also cited a few new papers and had some discussions to ensure the reliability of ORCTM for the internal solitary wave simulation. The content changes in the manuscript are highlighted with red color in revision mode, except for the English proofreading and corrections, and the equations and line numbers in the letter refer to the new version of the revised manuscript. The main comments and issues raised by the referee are listed below.

1. ***More details about the setup of the numerical test cases should be supplemented. For example, the bottom friction coefficients and the viscous coefficients are missing for test cases 3.2-3.5.***

**Answer**: Thanks. We checked all numerical test cases and have added more details about their associated model configurations, including bottom friction and viscous coefficients.

2. ***The boundary condition used to enforce the tidal flow in test case 3.5 should be added, because the internal waves are generated by the tidal flow in this test case.***

**Answer**: Thanks for pointing out the critical information we should have focused on. We have added a clear explication on tidal open boundary conditions in Exp. 3.5. Also, the time relaxation coefficient and sponge layer thickness at the west and east open boundaries have been supplemented and noted.

3. ***Two internal wave models (Ai and Ding, 2016; Ai et al., 2021) similar to the proposed model should be mentioned in the introduction section. The model developed by Ai and Ding (2016) employed a novel grid arrangement based on a 3D grid system, which is built from horizontally unstructured triangular grids and a vertical boundary-fitted coordinate system. The resulting Poisson equation has a symmetric sparse matrix. The model developed by Ai et al. (2021) incorporates the immersed boundary method to deal with the uneven bottoms for internal wave generations and propagations, which avoids numerical errors in the calculation of the baroclinic pressure force.***

**Answer:** Thanks for providing these helpful references. We have cited the two new nonhydrostatic model development papers in the introduction section and reviewed their numerical method for bottom boundary condition. The boundary conditions matched with the whole nonhydrostatic

algorithm are expected to determine a good performance of complex nonhydrostatic dynamics in the regional ocean model. Besides, we also compared our model with other related nonhydrostatic model development and evaluated the performance of ORCTM in the final section of the revised manuscript.

**Response letter to the referee #2**

*The paper describes the non-hydrostatic Oceanic Circulation and Tide Model (ORCTM) and numerical experiments on reproducing internal solitary wave dynamics with different types of analytical topography. The authors concluded that the developed model adequately simulates the behavior and evolution of the non-linear internal waves and tide-topography interaction. However, I found that the manuscript in its present form includes numerous flaws and needs to be substantially revised.*

We express the most gratitude to the referee for raising these important questions and comments. In the revision, we have added more details on the numerical test cases to clarify the validation experiment configurations. We also cited a few new papers and had some discussions to ensure the reliability of ORCTM for the internal solitary wave simulation. The content changes in the manuscript are highlighted with red color in revision mode, except for the English proofreading and corrections, and the equations and line numbers in the letter refer to the new version of the revised manuscript. The main comments and issues raised by the referee are listed below.

*General comments:*

*1) The model description is inaccurate in many places and requires deep revision. Specifically,*
*- Eqs. (4) and (5) are not independent and (4) is just the form of the continuity equation (3) integrated over the vertical coordinate with a kinematic boundary condition on top.*

**Answer**: Thanks for pointing out this untenable manifestation. In the model description section, we checked these equations and have rewritten the ocean primitive equations $(1) - (7)$ and the free surface elevation equation (8) separately.

*- The authors use the splitting of the total pressure for components, however it is unclear what components they use and how this splitting was performed. It becomes more clear only with eqs. (15)-(16) in the middle of Sect. 2.2, which is confusing.*

**Answer**: Thanks for this comment. We have rectified the expression of the hydrostatic balance equation (18) and carefully added more details on the total pressure splitting. The vertical momentum equation is updated into the nonhydrostatic balance equation (19), where the vertical velocity is controlled by the nonhydrostatic pressure gradient force and other terms.

*- The Boussinesq approximation does not assume the reference density in (3).*

**Answer**: Thanks for pointing out this mistake. We have modified this invalid expression.

*- Eq.(8) describes water density which is not used somewhere in (1)-(7).*

**Answer**: Thanks for pointing out this careless mistake. We have rectified the hydrostatic balance equation (18) so that the water density can be used in the hydrostatic pressure calculation.

*- The authors should explicitly formulate what boundary conditions they use at the open boundaries of the model domain. The same goes for all numerical experiments described in Section 3.*

**Answer**: Thanks for pointing out the critical information we should have focused on. We have added a clear explanation on tidal boundary conditions in Exp. 3.5, and the related coefficients have also been supplemented. The other numerical experiments are carried in a closed tank without the open boundary condition. Moreover, we have added the specific expression about the boundary conditions in the model description section.

*2) Despite a fair description of the model, the benefits of using ORCTM compared to other non-hydrostatic models cited in the ms are entirely unclear. What is the novelty of this model? I would love to see an additional paragraph or two where the authors would specify the novelty in the physical formulation of the model or boundary conditions used and/or in numerical implementation. In their present form, both these sides look quite trivial, which raises the question about the target for this paper. Do we need just one more non-hydrostatic model?*

**Answer:** Thanks for raising these meaningful comments and issues. We have supplied more detailed comparisons with other models and added the related discussion into the final section of the revised text in the Line 662–685.

Compared to other nonhydrostatic models, ORCTM is based on the finite difference method and owns a Z-coordinate. Although our validation experiments are indeed similar to other models, our model can effectively avoid some numerical errors existing in other models. For instance, Berntsen et al. (2006) indicated some noisy structures due to numerical errors of finite volume treatment near the bottom layer, so a high-order filter is needed to suppress this noise. In addition, the artificial flow usually emerges and has a negative impact on the ISWs breaking simulation because of the internal baroclinic pressure errors in the sigma-coordinate. Another advantage of ORCTM is the usage of the orthogonal curvilinear mesh grid. It can be employed to make the small-scale nonhydrostatic dynamics well-resolved in the concerned region via mesh refinement and construct reliable external boundary conditions via nest technique. All these advantages and validation experiments demonstrate that ORCTM can approach or reach an acceptable better level of the nonhydrostatic ocean model for the ISWs simulation.

*3) It was nice to see in Section 3 that the model can adequately reproduce the physics of the solitary waves under conditions of different topography. The authors did a good job presenting these results. However, a direct comparison of the model*

*results with analytical solutions for the most uncomplicated idealized cases is missing. In the case of lack of comparison with direct observations, this is the only way how we can attest the robustness of the model. I believe that this addition would benefit the manuscript.*

*Additionally, over the entire manuscript, I found lots of language mistakes and incorrect terms that make the text hard to read and understand. As a result, in many places, I cannot follow what the authors are trying to say. Therefore, I would strongly suggest the authors do professional proofreading of their text before the next iteration.*

**Answer:** Thanks for these comments and suggestions. It is a tradition for the model assessment for making comparisons with direct observations, but observational data are rare and precious. The reason why we have been wanting to develop a nonhydrostatic model is because the numerical model can mirror the macroscopic structure and assist with the implementation of observations for ISWs study.

It is noticed that our test cases are primarily based on other mature model test configurations (Bourgault and Kelley, 2004; Lai et al., 2010) which are derived from previous direct observations. For instance, After Michallet and Ivey (1999, called M&I) carried out the ISW experiment in the laboratory to study the velocity fields structure of wave breaking, the majority of nonhydrostatic model assessments are based M&I to construct the numerical ISW experiments (Bourgault and Kelley, 2004; Berntsen et al., 2006; Lai et al., 2010). The lock-exhange problem is also the same way. The agreement with the direct observations from M&I has already proven attest the robustness of the model as well as the comparisons with the theoretical models.

We also carry out the Exp. 3.5 modeled on the control run from Li (2010) and Zhang et al. (2011) to reproduce the major structures of nonlinear internal waves in the South China Sea. Because their 2-D or 3-D experiments can be considered as reliable and acceptable numerical predictions that have already been compared well with the most in-situ observations, such as the structure and timing of type-a and b waves in the South China Sea (Ramp et al., 2019). Therefore, we think our comparison results have insinuated that the ISW simulations via using ORCTM agree well with the realistic observational characteristics.

*Specific comments:*
*L10: How could the boundary conditions support the regional simulations?*
*L12: The equations cannot consider something. Check the language.*
*L37-40: Unclear sentence. Reformulate, please.*
*L44-47: How could the hydrostatic approximation result in the inapplicability of the nonhydrostatic dynamics? Clarify.*

**Answer:** Thanks for raising these questions. We have added more explanation in the Line 10–47 of the revised text.

*L89-95: The sentence is hard to understand. Try to split it to make clearer.*
*L103: The Cartesian coordinate system does not require the eastward and northward velocity components.*

**Answer:** Thanks. We have revised the expression into the local cartesian framework of reference on the rotating earth in the Line 112-114 of the revised text.

*L111: Check the components of the forcing vector.*
*L112: Usually, the river runoff and boundary inflow are specified through the boundary conditions but not in the model equations. How are the authors supposed to adjust such a forcing with the boundary conditions?*
*L115: Check the components of the viscosity vector.*
*L125-127: Provide exact equations for boundary conditions.*
*Eq.(16). Not sure that the reference density is correct here. Otherwise, it needs more details.*
*L191-193: Unclear sentence. Please, reformulate.*

**Answer:** Thanks. We have revised these ambiguous and invalid expressions and added the exact expression formulas of boundary conditions in the Line 117–141 of the revised text.

*Figure 1a: This is a density anomaly, not just density.*
*L210-213: Unclear meaning.*
*L241: You already introduced the free surface elevation in L104.*

**Answer:** we have defined $\sigma = \rho - 1000$ kg m$^{-3}$ in Line 343 of the revised text.

*L266: Put both these densities in Fig. 4*
*Figure.6 : Use (a), (b) and (c) notation in the title.*

**Answer:** We have added the water density information in Fig.4 and corrected these notations of the title in Fig.6 based on the publishment of GMD.

*L311: Check the language.*
*L332: What does the "rear effect" mean?*

Answer: We have revised this expression in the Line 347–349.

*Figure 16: The left panels are unreadable.*

**Answer:** These left snapshots are Fig. 4 from Michallet and Ivey (1999) which are grayscale images. We have upgraded a clearer picture based on their original paper.

*L466: How can the horizontally uniform stratification drive the model?*

Answer: We have corrected it to the expression that the horizontally uniform stratification is used as the initial field for our model in the Line 494–495 of the revised text.

***Appendix A: Check the correctness of indices in A.2-A.4. It is unclear how were Au, Av, and Aw calculated. Please, reformulate using DX, DY, and DZ notation.***

**Answer:** Following the suggestion. We have added these equation expressions using DX, DY, and DZ notation in A.5, which will be a clear illustration there in the Line 705–706 of revised text.

***L678: What does "harmony" mean here? Please, clarify.***

**Answer**: We develop this nonhydrostatic regional ocean model as an extension of original hydrostatic model. If the flow is close to nonhydrostatic, the nonhydrostatic module will reproduce this flow and the hydrostatic balance module will not. However, the hydrostatic solution of this flow is still obtained by the hydrostatic module. The nonhydrostatic dynamics module will fill in the deficiency of hydrostatic module. In other words, the nonhydrostatic and hydrostatic dynamic modules should be independent of each other and not contradictory, which means the nonhydrostatic and hydrostatic components can be simultaneous in the model.

***In summary, based on my evaluation of the current version of the manuscript, I would suggest major revision.***

**Answer**: We have taken the referee's all suggestions and refined carefully the English expression in the new submission.

**References**

Ai. C., and Ding., W.: A 3D unstructured non-hydrostatic ocean model for internal waves: Ocean Dyn, 66, 1253-1270, https://doi.org/ 10.1007/s10236-016-0980-9, 2016.

Ai. C., Ma. Y., Yuan. C., and Dong, G.: Non-hydrostatic model for internal wave generations and propagations using immersed boundary method, Ocean. Eng., 225, 108801, https://doi.org/ 10.1016/j.oceaneng.2021.108801, 2021.

Bourgault, D. and Kelley, D. E.: A laterally averaged nonhydrostatic ocean model, J. Atmos. Ocean. Tech., 21, 1910-1924, https://doi.org/10.1175/JTECH-1674.1, 2004.

Berntsen, J., Xing, J., and Alendal, G.: Assessment of non-hydrostatic ocean models using laboratory scale problems, Cont. Shelf. Res., 26, 1433-1447, https://doi.org/10.1016/j.csr.2006.02.014, 2006.

Li, Q.: Numerical assessment of factors affecting nonlinear internal waves in the South China Sea, Prog. Oceanogr., 121, 24-43, http://dx.doi.org/10.1016/j.pocean.2013.03.006, 2014.

Lai, Z., Chen, C., Cowles, G. W., and Beardsley, R. C.: A nonhydrostatic version of FVCOM: 1. Validation experiments, J. Geophys. Res.-Ocean, 115, C11010, https://doi.org/10.1029/2009JC005525, 2010.

Michallet, H., and Ivey, G. N.: Experiments on mixing due to internal solitary waves breaking on uniform slopes, J. Geophys. Res.-Ocean, 104, 13467-13477, https://doi.org/10.1029/1999JC900037, 1999.

Ramp, S. R., Park, J. -H., Yang, Y. J., Bahr, F. L., and Jeon, C.: Latitudinal Structure of Solitons in the South China Sea, J. Phys. Oceanogr., 49, 1747-1767, https://doi.org/10.1175/JPO-D-18-0071.1, 2019.

Zhang, Z., Fringer, O. B., and Ramp, S. R.: Three-dimensional, nonhydrostatic numerical simulation of nonlinear internal wave generation and propagation in the South China Sea, J. Geophys. Res.-Ocean, 116, C05022, https://doi.org/10.1029/2010JC006424, 2011.